# Large Language Model-guided Multi-modal Motion Planning via Mixed Integer Program

## Abstract

Multi-Modal Motion Planning ($M^3P$) is a rather challenging form of motion planning where the planner searches through the continuous space of motions as well as discrete space of modes. For instance, a biped robot may need to walk to a target location and then use its arms to grasp an object, capturing both mode transitions and continuous dynamics to find feasible paths that neither purely discrete nor continuous planners can handle. However, brute-force global search is typically sample-inefficient and computationally expensive. Recent research has explored the use of Mixed-Integer Programming (MIP) to reformulate global search problems in robotic applications. MIP leverages the branch-and-bound algorithm to efficiently prune infeasible or sub-optimal solutions. Despite its strengths, MIP is limited to problems with disjoint convex feasible domains—a constraint that is often too restrictive for general motion planning. To address this, prior work has proposed techniques to approximate non-convex motion planning problems as disjoint convex MIPs. Unfortunately, these methods are typically hand-crafted and domain-specific, limiting their generalizability. In this work, we explore the use of Large Language Models (LLMs) to automatically translate non-convex optimization problems into approximate MIP formulations. To this end, we construct a dataset comprising various $M^3P$ problems paired with their known MIP approximations. We evaluate LLM performance on this reformulation task using both In-Context Learning (ICL) and Supervised Fine-Tuning (SFT). Our results demonstrate that LLMs are capable of capturing common patterns in MIP reformulations and can even generalize to complex, unseen translation tasks beyond those encountered during fine-tuning.

## 1 Introduction

In this paper, we address the problem of Multi-modal Motion Planning ($M^3P$), originally proposed in (Hauser et al., 2010; Hauser & Latombe, 2010). $M^3P$ extends classical motion planning by allowing a robot to transition between fundamentally different modes of operation—for example, switching between grasping and non-grasping states, or navigating via driving versus flying. These discrete mode switches introduce significant complexity, but they are essential for solving a wide range of real-world robotics problems, including manipulation, locomotion, and aerial-ground co-ordination. As such, $M^3P$ is a core component in enabling intelligent, multi-capable robotic systems to operate in complex environments. $M^3P$ can be viewed as a specialized subproblem within the broader framework of Task and Motion Planning (TAMP), which involves reasoning over both high-level discrete decisions and low-level continuous motions. Like TAMP, $M^3P$ requires solving challenging search problems in hybrid decision spaces that span both discrete and continuous domains. The interleaving of symbolic decisions (e.g., choosing modes) with geometric constraints (e.g., feasible trajectories) creates a large, often intractable, planning space. Recent years have seen significant advances in both general-purpose TAMP (Dantam et al., 2016; Toussaint et al., 2018; Garrett et al., 2021; Jiao et al., 2022) and $M^3P$ (Kingston et al., 2020; Beyer et al., 2021; Kingston & Kavraki, 2022), particularly with respect to improved scalability and planning efficiency. However, many of these approaches rely heavily on domain-specific heuristics, hand-tuned parameters, or environment-specific assumptions, limiting their applicability and generalizability to diverse robotic settings. Moreover, a number of state-of-the-art methods are built upon assumptions such as local optimality (Zhao et al., 2024; Manchester & Kuindersma, 2019) or differentiability of the planning

landscape (Toussaint et al., 2018; Envall et al., 2023). While these assumptions can simplify algorithm design and accelerate computation, they often do not hold in practice—especially in cluttered, discontinuous, or highly constrained environments.

Given the limitations of traditional TAMP, especially the weak coupling between task and motion planning, researchers have increasingly explored unified formulations and integrated solvers. One approach involves using the Planning Domain Definition Language (PDDL), a high-level formalism for specifying symbolic planning problems, along with powerful domain-independent search algorithms such as Fast-Downward (Helmert, 2006) and Nyx (Piotrowski et al., 2024). These symbolic planners are capable of efficiently generating high-level action sequences in complex decision spaces. However, PDDL-based planners operate solely on symbolic abstractions and do not account for geometric feasibility or generate executable low-level motion plans. As a result, TAMP systems must interface symbolic planners with separate motion planners (Garrett et al., 2021), often through heuristics or manually engineered bridging layers. This loose integration can lead to inefficiencies, inconsistencies, or planning failures, especially in domains where symbolic actions tightly depend on geometric feasibility. To address this disconnect, a growing body of research has proposed the use of Mixed-Integer Programming (MIP) to directly model and solve certain subclasses of $M^3P$ problems. MIP-based approaches have been successfully applied to tasks such as grasp synthesis (Liu et al., 2020), caging (Aceituno-Cabezas et al., 2019), inverse kinematics (Dai et al., 2019), multi-agent and UAV motion planning (Deits & Tedrake, 2015b; Yu & LaValle, 2013; Marcucci et al., 2023), and footstep planning (Deits & Tedrake, 2014). These formulations allow for unified reasoning over both discrete decisions (mode switches, action choices) and continuous variables (robot poses, trajectories), thereby enabling a more tightly integrated solution to $M^3P$ tasks. Compared to PDDL-based symbolic search, MIP offers several key advantages. Most notably, MIP enables joint optimization over hybrid decision spaces, capturing task–motion dependencies within a single formulation. Additionally, decades of advances in operations research, such as heuristics and branch-and-bound strategies, make MIP solvers highly efficient, often surpassing brute-force sampling and decoupled planning.

With the rapid success of Large Language Models (LLMs), their influence has extended far beyond natural language processing, significantly impacting robotic system design and human-robot interaction. A growing body of research has explored the use of LLMs for motion planning tasks (Ahn et al., 2022; Ding et al., 2023; Driess et al., 2023; Huang et al., 2022), leveraging their ability to interpret and reason over high-level task descriptions. While such approaches have demonstrated promising results, the long-horizon reasoning capabilities of LLMs remain limited. To address these limitations, a more recent paradigm (LLM+P) (Liu et al., 2023; Wang et al., 2024; Silver et al., 2024) has emerged. In this framework, the LLM is used as a translator—converting natural language problem descriptions into PDDL scripts. These symbolic plans are then executed by conventional task planners, leveraging the strengths of both systems: the language understanding capabilities of LLMs and the search efficiency of symbolic planners. Building on the success of LLM+P, we pose the question: *Can LLMs be used to generate MIP problem descriptions directly from natural language?* We refer to this emerging paradigm as LLM+MIP. The proposed LLM+MIP framework offers several notable advantages. First, similar to LLM+P, it combines the flexibility and generalization abilities of LLMs in language translation with the robustness and optimality guarantees of classical optimization solvers. Second, and more importantly, unlike symbolic planners that operate purely over discrete action spaces, MIP-based solvers enable joint reasoning over both discrete and continuous variables, directly generating solutions to $M^3P$ problems.

While the LLM+MIP paradigm offers exciting new possibilities, it also introduces a set of significantly more complex challenges compared to the LLM+P approach. In the case of LLM+P, the task is primarily one of text-to-text translation. In contrast, translating natural language into MIP solver scripts is a much more demanding task, as it requires not only linguistic understanding but also mathematical reasoning and geometric abstraction. MIP can only solve a specific class of non-convex optimization problems—those where the feasible region is disjointly convex, meaning it can be expressed as a finite union of convex subsets. However, most real-world robotic motion planning problems exhibit generally non-convex feasible domains. To make such problems tractable for MIP solvers, it is necessary to approximate the original non-convex feasible space using discretization techniques that transform the continuous, non-convex problem into a finite collection of convex subregions (Sherali & Wang, 2001; Deits & Tedrake, 2015a; Amice et al., 2022). However, the selection of appropriate discretization methods, parameters, and constraints is highly problem-specific and re-

quires deep mathematical insight. Poor choices can lead to infeasible formulations or dramatically degrade solver performance. Given recent advances in the mathematical reasoning capabilities of LLM (Yu et al., 2025; Xiong et al., 2024; Shi et al., 2024), we hypothesize that "LLMs are capable of automatically generating MIP solver scripts from natural language descriptions of $M^3P$ problems".

To explore this hypothesis, we introduce a novel benchmark dataset and evaluation framework for assessing the capability of LLMs to perform LLM+MIP translation. Our dataset consists of a variety of 2D single- and multi-domain $M^3P$ instances, covering both basic and more complex scenarios. To reduce implementation complexity and isolate the reasoning task, we abstract discretization strategies as callable API functions, allowing the LLM to focus on selecting and composing these building blocks based on problem context. Our experiments demonstrate that LLMs can successfully generate MIP solver scripts for many instances in both single- and multi-domain $M^3P$ tasks. Furthermore, we show that the success rate improves significantly when the model is fine-tuned on domain-specific training data, highlighting the potential of specialization in enabling robust performance. We believe this work opens up a new research direction at the intersection of natural language understanding, mathematical programming, and robot motion planning. The LLM+MIP paradigm represents a step toward end-to-end robotic systems that can reason over high-level task descriptions and produce executable, optimization-based plans. **To summarize**, we are the first to fine-tune LLMs for $M^3P$, extending beyond symbolic action (LLM+P) to integrate symbolic with continuous planning, achieving success rates of 98.8% on single-domain and 89.6% on multi-domain tasks.

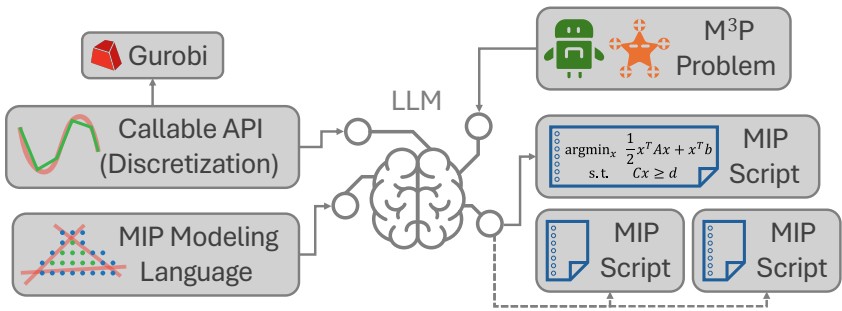

**Figure 1:** Illustration of LLM+MIP paradigm: LLM has access to both MIP modeling language (CVXPY in our case with Gurobi backend), as well as an additional set of API for access to discretization methods. Given an $M^3P$ instance in natural language, LLM generates the MIP script for solving the problem. It can also decompose the problem into smaller, more tractable MIPs (dashed lines).

## 2 RELATED WORK

We review related work on TAMP, $M^3P$, MIP, and the recent progress on the combination of LLM and conventional search algorithms. Extended discussions are provided in Section E.

**TAMP and $M^3P$:** TAMP (Garrett et al., 2021; Dantam et al., 2016) represents a fundamental paradigm in robotics, as it enables a robot to reason over long-horizon symbolic action sequences in order to accomplish complex, high-level tasks. Within this broader TAMP framework, $M^3P$ focuses on a particular subclass of problems where a robot must navigate between different modes of locomotion or contact in order to complete a task (Hauser et al., 2010; Hauser & Latombe, 2010). A growing body of work has explored MIP approaches to $M^3P$ (Liu et al., 2020; Aceituno-Cabezas et al., 2019; Dai et al., 2019; Deits & Tedrake, 2015b; Yu & LaValle, 2013; Marcucci et al., 2023; Deits & Tedrake, 2014), which preserve the unified representation of modes while exploiting the branch-and-bound search paradigm inherent to modern MIP solvers.

**MIP & Data-driven Methods:** MIP formulations can be divided into exact and approximate Klose & Drexl (2005); Xiong et al. (2022); Braekers et al. (2016); Sherali & Wang (2001); Deits & Tedrake (2015a); Amice et al. (2022). Formulating approximate MIP is substantially more challenging: because discretization is problem-dependent and requires carefully chosen strategies to balance tractability and fidelity. This dual nature makes approximate MIP both flexible and demanding. While approximate MIP only provides approximate solutions, it has proven effective for many difficult $M^3P$ instances. Recent research has explored the integration of data-driven acceleration techniques, such as learned branching heuristics (Zhang et al., 2024), node selection (Labassi et al., 2022), or cutting planes (Wang et al., 2023). Recently, LLMs have been incorporated into this workflow (Li et al., 2023a; Chen et al., 2024; Liu et al., 2024), including translating problem descriptions into MIP scripts Li et al. (2023b); AhmadiTeshnizi et al. (2024), building a foundation

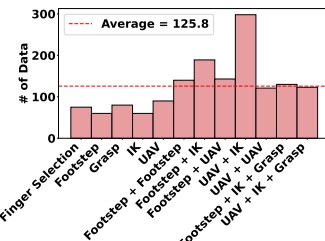
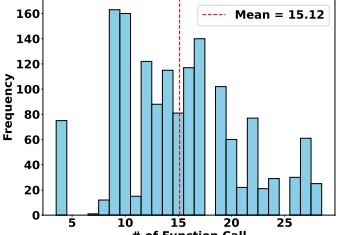
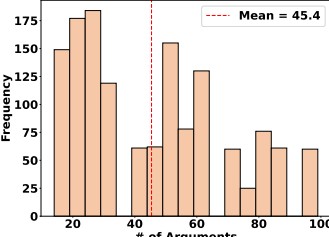

Figure 2: Task dist.  Figure 3: Function calls dist.  Figure 4: #arguments dist.

Figure 5: Statistics of the RoboM$^3$P dataset. (a) The distribution of data across twelve task categories, averaging 125.8 samples per category, with both single and multi-task compositions. (b) The number of function calls per generated MIP code (mean = 15.1). (c) The number of arguments per trajectory (mean = 45.4).

model to efficiently solve MILP Li et al. (2025). However, these methods focus exclusively on exact MIP. Similarly, we are aware of several MIP datasets (Prouvost et al., 2020) for benchmarking performances. However, all these datasets contain only exact instances. In contrast, our work targets approximate MIP, where the LLM must reason mathematically about discretization choices. To the best of our knowledge, our dataset is the first one for benchmarking approximate MIP instances with a focus on M$^3$P applications.

**LLM-as-Planner and LLM-with-Planner:** LLMs have inspired research on solving TAMP through prompt engineering, a paradigm referred to as LLM-as-Planner (Ahn et al., 2022; Ding et al., 2023; Driess et al., 2023; Huang et al., 2022). To overcome long-horizon reasoning limitations, a complementary paradigm called LLM-with-Planner has been proposed (Liu et al., 2023), where the LLM converts natural-language descriptions into PDDL and conventional planners solve them. Subsequent works have expanded this paradigm with interfaces to motion planners (Wang et al., 2024), domain generalization (Silver et al., 2024), and structured memory (Agarwal et al., 2025). In this work, we push this line further by introducing LLM+MIP, a hybrid paradigm in which the LLM generates MIP formulations, providing a unified mathematical framework that can simultaneously capture discrete decisions and continuous optimization variables specifically for robotic tasks. While our focus is on unimodal (text-based) inputs, we note that a separate line of work has explored the integration of multi-modal features into robotic foundation models like OpenVLA (Kim et al., 2024) which incorporates visual grounding. Such extensions are orthogonal to our contributions, though we view incorporating visual descriptions as a promising direction for future research.

## 3 METHOD

Our overall pipeline is illustrated in Figure 1. To enable an LLM to translate M$^3$P problem descriptions into executable MIP formulations, we treat an existing MIP modeling language as the LLM-callable API. In this work, we adopt CVXPY (Diamond & Boyd, 2016) as our modeling interface. CVXPY is a widely used high-level optimization package while supporting multiple backend solvers. For all experiments, we use Gurobi (Pedroso, 2011) as the default backend solvers. As discussed earlier, solving inexact MIP formulations for M$^3$P requires discretization techniques (Sherali & Wang, 2001; Deits & Tedrake, 2015a; Amice et al., 2022) that approximate generally non-convex constraints with disjoint convex representations. However, asking an LLM to autonomously design or derive these discretization schemes is beyond the scope of this work. Instead, we provide a library of discretization primitives, implemented as LLM-callable APIs alongside the CVXPY modeling interface. In Section 3.1, we introduce our dataset, which pairs a set of natural-language 2D M$^3$P problem descriptions with their corresponding MIP formulations. In Section 3.2, we present the set of discretization techniques implemented in our system and explain how they are exposed as callable APIs. Finally, in Section 3.3, we describe our LLM+MIP translation pipeline.

### 3.1 ROBOM$^3$P DATASET

We introduce RoboM$^3$P, a dataset of 1500 sampled M$^3$P problems specifically designed to evaluate the capability of LLM+MIP frameworks. Each sample in the dataset contains three core

components: (i) a natural-language description of the task, (ii) an obstacle map that specifies the environment layout and object boundaries, and (iii) the final MIP code encoding feasible trajectories. The MIP code is expressed as a structured sequence of function-like API calls, for example: `xxx_constraint(arg1=val1)`. In addition to the core modeling APIs provided by CVXPY, we include a set of LLM-callable discretization APIs, along with their detailed semantics, in Section B, which encapsulate common techniques for approximating non-convex or otherwise complex constraints into tractable collections of linear or convex constraints. Detailed examples of these data pairs, are provided in the Section D.2, Section D.3, Section D.5. Together, these calls define the optimization problem by specifying decision variables and constraints, thereby representing full motion plans in a structured and executable form. The dataset spans five single-domain $M^3P$ tasks, as well as seven multi-domain compositions that integrate locomotion, manipulation, and aerial navigation as detailed in Section A. This breadth allows RoboM$^3$P to capture both the modularity of individual planning problems and the compounded difficulty of multi-modal coordination. A statistical overview of the dataset is provided in Figure 5. Panel (a) shows the distribution across the twelve task categories, with a mean of 125.8 samples per category. The dataset is relatively balanced across single-domain tasks, while multi-domain compositions such as UAV+IK are more heavily represented due to our extensive testing of spatial constraints in those settings. Panel (b) reports the distribution of function calls per generated program, with an average of 15.1 calls per sample, reflecting the non-trivial size of the generated formulations. Panel (c) presents the distribution of arguments used across trajectories, averaging 45.4 arguments per instance, which highlights the structural and combinatorial complexity of the encoded optimization problems. Taken together, these statistics demonstrate that RoboM$^3$P is both diverse and structurally rich, providing a challenging benchmark for studying how well LLMs can generalize to generate correct, executable, and scalable MIP formulations for robotic motion planning tasks. Finally, we emphasize that although all problems in RoboM$^3$P are defined in a 2D workspace, each of them naturally extends to 3D variants with corresponding MIP relaxations and to more complex compositions involving larger numbers of robots (e.g., coordinating five UAVs and five bipedal robots with grippers to grasp ten objects). In this work, however, we restrict our evaluation to at most two-task compositions in 2D primarily for efficiency, as these setting formulations requires significantly less computational time from the MIP solver while still preserving the essential challenges of the tasks.

## 3.2 LLM-Callable API & Discretization Techniques

MIP solves convex optimization problems in which some decision variables are restricted to take integer values. As a result, MIP is only directly applicable when the feasible region of a problem can be expressed as a finite union of convex sets. However, many of the problems in RoboM$^3$P involve inherently non-convex constraints—for example, collision avoidance, kinematic feasibility, and trajectory smoothness—which cannot be expressed in this form. To address this gap, we implement a suite of discretization methods that approximate general non-convex constraints as collections of disjoint convex regions. These methods are exposed through a set of LLM-callable APIs, which are summarized in Section B. Due to page limit, we put detailed mathematic formulation in Section C. Broadly, the APIs fall into three main categories.

**Collision-Free Constraints:** The most critical class of APIs deals with collision avoidance. We implement the IRIS algorithm (Deits & Tedrake, 2015a), which partitions a non-convex collision-free domain into a set of convex regions (IRIS regions). This functionality is provided through the `create_map` API, which generates convex regions for use in planning problems. Such regions are required across multiple tasks: footsteps must lie in collision-free regions for locomotion, UAV waypoints must be placed in obstacle-free corridors, and articulated links in IK problems must avoid collisions. Beyond the vanilla IRIS method, **we introduce novel variants that allow users to encode additional spatial requirements: `add_spatial_relation_constraints`. We describe the detailed formulation in Section C.2.** For example, to require a UAV to circle around an object, we generate four IRIS regions placed sequentially around the obstacle and enforce order-dependent constraints. Similarly, users can require UAV trajectories to pass from the "left" or "right" side of an obstacle by constructing directional IRIS regions and binding them through additional ordering constraints. Such spatial constraints can also be extended to other tasks, such as IK. We demonstrate the spatial result on both UAV and IK in Figure 6.

**Trajectory Constraints:** For UAVs, the planned trajectory must not only remain collision-free but also maintain smoothness over time. To capture this, we parameterize UAV paths as spline curves

and introduce constraints on their control points. Collision-free regions are enforced at the spline level via `add_control_points_constraints`, while smooth connectivity across curve segments is ensured with `add_continuity_constraints`. For footstep tasks, higher-level behavioral requirements—such as enforcing left/right ordering and determine footstep reachable area—can be maintained via monotonicity and polynomial approximation, implemented through the `add_monotonicity_constraints` and `add_reachability_constraints` API. These constraints build on well-established discretization and relaxation techniques for non-convex trajectory optimization Sherali & Wang (2001).

**IK and Grasp Constraints:** Manipulation problems introduce a different source of non-convexity: the trigonometric sin and cos functions that describe revolute joint kinematics. To approximate these nonlinearities, we discretize the 2D rotation space using piecewise-linear approximations, following methods from Deits & Tedrake (2014). Such constraints are implemented through `add_chain_rotation_constraints`. For grasping, we adopt the pipeline of (Liu et al., 2020), which first samples potential grasp contact points on an object's surface via `get_object_surface_samples`. The inverse kinematics solver is then constrained to reach one of these points using `add_end_constraints`. Finally, grasp quality is enforced by discretizing grasp wrench metrics into an optimization objective, specified through the `add_grasp_wrench_constraints` API. This enables the LLM+MIP framework to reason not only about kinematic feasibility but also about the relative stability of different grasp candidates.

### 3.3 LLM+MIP Procedure

Due to the inherent challenges of combining LLMs with MIP, we explore two approaches for problem translation: zero-shot prompting and supervised finetuning (SFT). This contrasts with LLM+P (Liu et al., 2023), which relies solely on zero-shot translation. Within zero-shot prompting, we evaluate two variants of in-context learning (ICL): input-output (IO) prompting and chain-of-thought (CoT) (Wei et al., 2022) prompting. For both IO and CoT, the LLM is provided with five input-output examples drawn from single-domain tasks. In the IO setting, the LLM is expected to directly generate the MIP script. In the CoT setting, we first prompt the LLM to classify the task type (one of five single-domain or seven multi-domain tasks), and then prompt it to produce the corresponding MIP script. While more advanced ICL methods such as tree of thoughts (Yao et al., 2023) and graph of thoughts (Besta et al., 2024) exist, applying them in our setting is impractical due to the difficulty of defining suitable thought generators and evaluators. Beyond ICL, we also assess the effectiveness of SFT, where the LLM is finetuned on a subset of single- or multi-domain tasks and tested on previously unseen tasks.

We further observe that the computational cost of solving MIP grows superlinearly with problem size, making large instances particularly expensive. A practical approach to mitigating this cost is to decompose the original MIP into smaller subproblems, each of which can be solved much more efficiently. This strategy is especially effective for multi-domain tasks. For instance, in the Footstep+UAV task, the bipedal robot and the UAV are required to meet at a common location. If the LLM identifies this meeting point, the task can be decomposed into two independent problems: a Footstep task for the robot and a UAV task for the drone, each directing the agent to its designated goal. Motivated by this, we propose using the LLM as a MIP optimizer to decompose multi-domain $M^3P$ tasks into smaller, more tractable MIP scripts, as illustrated in Figure 1 (dashed line).

## 4 Evaluation

Our evaluation focuses on two central questions: **(i)** *Can LLMs correctly use the provided API functions to assemble a single MIP formulation that directly solves an $M^3P$ task described in natural language?* **(ii)** *Can LLMs decompose a multi-domain $M^3P$ task into single-domain subproblems?*

To answer these questions, we adopt three complementary evaluation metrics: **Formulation-Correctness:** whether the LLM identifies the correct set of MIP constraints, indicating its understanding of which constraints are necessary to model the task. **Parameter-Correctness:** whether the LLM accurately maps text-specified requirements (e.g., start/goal positions, reachability bounds) into the correct function arguments. **Success-Rate:** whether the resulting MIP problem can be exe-

cuted by a solver to produce a feasible plan, thereby testing the end-to-end ability to translate natural language into solvable MIP problems.

We compare three training regimes: **ICL**, **SFT-Single**, and **SFT-Full**. **ICL** is training-free, using five in-context input–output pairs drawn from five different single-domain tasks. **SFT-Single** finetunes the model on 250 examples sampled from five single-domain task datasets (50 per task). **SFT-Full** extends this training by including an additional 490 examples from multi-domain tasks (70 per task), covering all 12 task types. Evaluation is performed on held-out datapoints to ensure a strict train–test split. Unless otherwise specified, GPT-4o serves as the base LLM. Gurobi-based MIP solvers solve single-domain tasks within 60 s and multi-domain tasks in about 300 s on average on MacOS system with Apple M1 CPU, offering a reasonable computational cost for efficiently addressing M³P problems.

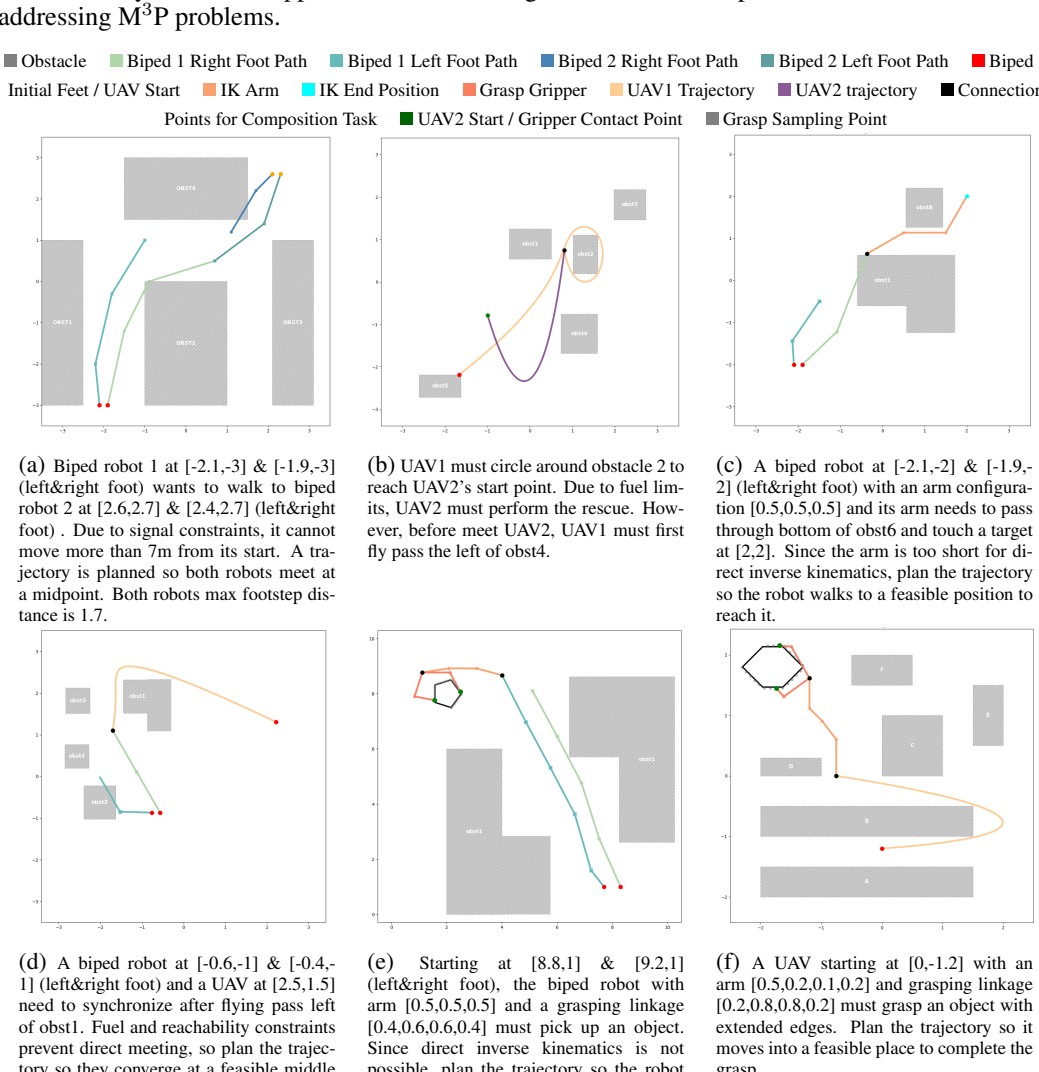

(a) Biped robot 1 at [-2.1,-3] & [-1.9,-3] (left&right foot) wants to walk to biped robot 2 at [2.6,2.7] & [2.4,2.7] (left&right foot) . Due to signal constraints, it cannot move more than 7m from its start. A trajectory is planned so both robots meet at a midpoint. Both robots max footstep distance is 1.7.

(b) UAV1 must circle around obstacle 2 to reach UAV2's start point. Due to fuel limits, UAV2 must perform the rescue. However, before meet UAV2, UAV1 must first fly pass the left of obst4.

(c) A biped robot at [-2.1,-2] & [-1.9,-2] (left&right foot) with an arm configuration [0.5,0.5,0.5] and its arm needs to pass through bottom of obst6 and touch a target at [2,2]. Since the arm is too short for direct inverse kinematics, plan the trajectory so the robot walks to a feasible position to reach it.

(d) A biped robot at [-0.6,-1] & [-0.4,-1] (left&right foot) and a UAV at [2.5,1.5] need to synchronize after flying pass left of obst1. Fuel and reachability constraints prevent direct meeting, so plan the trajectory so they converge at a feasible middle position.

(e) Starting at [8.8,1] & [9.2,1] (left&right foot), the biped robot with arm [0.5,0.5,0.5] and a grasping linkage [0.4,0.6,0.6,0.4] must pick up an object. Since direct inverse kinematics is not possible, plan the trajectory so the robot moves to a location where grasping is feasible.

(f) A UAV starting at [0,-1.2] with an arm [0.5,0.2,0.1,0.2] and grasping linkage [0.2,0.8,0.8,0.2] must grasp an object with extended edges. Plan the trajectory so it moves into a feasible place to complete the grasp.

Figure 6: Visual results of different types of M³P instances. The displayed trajectories are generated by executing code generated by SFT-fully finetuned LLM with the gurobi solver. The input text for each scenario is shown beneath the corresponding subfigure. For clarity, obstacle descriptions and grasp object geometries are omitted. The legend is provided at the top.

### 4.1 SINGLE UNIFIED MIP TRANSLATION

Our first experiment evaluates whether LLMs can translate a text description of an M³P instance into a single unified MIP problem. For example, in the UAV+Grasp task, the UAV position and the gripper's base location must coincide to reflect that the gripper is mounted on the UAV. Correctly

solving such multi-domain problems requires introducing additional constraints, e.g., by calling `share_point_feasibility_constraints`. Although these constraints are not explicitly specified in the text instructions, the LLM is expected to infer them from the problem description.

Results in Table 1 show that ICL often fails on complex tasks (as UAV tasks impose the most demanding spatial requirements, such as flying around, above, below, to the right, or to the left of specified regions), with solve rates as low as 4-42%. SFT-Single markedly improves correctness and solve rates on single-domain instances, but remains brittle when tasks are multi-domain. In contrast, SFT-Full achieves near-perfect performance across almost all tasks, including challenging multi-domain compositions such as UAV+Grasp. These findings demonstrate that finetuning is essential for enabling LLMs to compose heterogeneous motion-planning modalities into a coherent unified MIP formulation. We then compare different LLMs, both open-source and proprietary, and LLM

| Task | ICL (avg ± std) | SFT-Single (avg ± std) | SFT-Full (avg ± std) |
|---|---|---|---|
| Finger Selection | 100.0±0.0 / 100.0±0.0 / 100.0±0.0 | 100.0±0.0 / 100.0±0.0 / 100.0±0.0 | **100.0±0.0 / 100.0±0.0 / 100.0±0.0** |
| Footstep Planning | 100.0±0.0 / 92.0±1.2 / 89.6±0.3 | 100.0±0.0 / 100.0±0.0 / 100.0±0.0 | **100.0±0.0 / 100.0±0.0 / 100.0±0.0** |
| Grasp | 100.0±0.0 / 86.4±2.6 / 80.3±0.6 | 100.0±0.0 / 100.0±0.0 / 100.0±0.0 | **100.0±0.0 / 100.0±0.0 / 100.0±0.0** |
| IK | 88.9±0.0 / 83.3±0.0 / 70.9±0.6 | 100.0±0.0 / 100.0±0.0 / 92.9±0.6 | **100.0±0.0 / 100.0±0.0 / 95.4±2.6** |
| UAV | 49.4±3.9 / 55.9±2.9 / 42.4±4.3 | 93.7±0.0 / 94.9±0.0 / 89.1±0.3 | **100.0±0.0 / 100.0±0.0 / 98.5±0.9** |
| Footstep + Footstep | 47.4±0.0 / 52.5±0.0 / 35.3±5.3 | 52.6±0.0 / 52.5±0.0 / 40.0±2.9 | **100.0±0.0 / 100.0±0.0 / 100.0±0.0** |
| Footstep + IK | 56.2±0.0 / 52.4±0.0 / 45.3±2.8 | 59.6±0.0 / 82.5±0.0 / 57.1±5.2 | **100.0±0.0 / 100.0±0.0 / 100.0±0.0** |
| Footstep + UAV | 19.7±0.0 / 18.8±0.0 / 16.2±3.3 | 11.6±0.5 / 14.0±0.4 / 10.7±4.9 | **100.0±0.0 / 100.0±0.0 / 98.0±2.8** |
| UAV + IK | 46.7±0.0 / 61.3±0.0 / 22.3±1.2 | 46.7±0.0 / 61.3±0.0 / 22.3±1.2 | **80.0±0.0 / 83.9±0.0 / 80.0±0.0** |
| UAV + UAV | 28.9±1.4 / 38.8±1.0 / 25.4±1.3 | 40.4±1.6 / 51.6±1.5 / 36.7±3.6 | **100.0±0.0 / 100.0±0.0 / 99.2±0.0** |
| Footstep + Grasp | 34.6±0.0 / 56.0±0.0 / 30.4±0.7 | 34.6±0.0 / 56.0±0.0 / 30.4±0.7 | **94.9±8.9 / 97.8±3.9 / 92.6±12.8** |
| UAV + Grasp | 5.2±0.6 / 12.8±0.4 / 4.9±1.4 | 38.9±3.8 / 28.8±6.9 / 15.3±5.5 | **98.1±3.2 / 98.1±3.2 / 85.8±2.1** |

Table 1: LLM to directly solve the single- and multi-domain tasks as a single MIP problem. Comparison of Formulation-Correctness/Parameter-Correctness/Success-Rate(%) across ICL, SFT-Single, and SFT-Full.

specifically finetuned on math and code tasks, as our tasks inherently require complex reasoning and programming ability. We evaluate scenarios where the LLM is given only part of the API functions and must generate the remaining constraints using `cvxpy`. In the first setting, we remove simple calls (1-5 lines) such as `add_base_constraints` and `add_end_constraints`. The LLM reliably reproduces these constraints, achieving nearly the same Success Rate as with full APIs. In the second setting, we remove complex APIs (avg around 80 lines) such as `add_spatial_relation_constraints`, `add_chain_rotation_constraints`, `add_grasp_wrench_constraints`, and `add_reachability_constraints`, requiring the LLM to implement them from scratch. The Success Rate drops to zero, underscoring the difficulty of generating complex MIP constraints and the importance of high-level API abstractions. Formulation-Correctness is omitted since trajectories differ from other setups. Results in Table 2 highlight that SFT is critical for achieving robust generalization to multi-domain tasks, narrowing the gap between single-domain mastery and multi-domain integration. We also visualize the results in Figure 6. More various results are shown in Section F.

| Model | Single-Domain Task | | Multi-Domain Tasks | |
|---|---|---|---|---|
| | Formulation-Correctness ↑ | Success-Rate ↑ | Formulation-Correctness ↑ | Success-Rate ↑ |
| GPT-4o (partial API Easy) | - | 64.9 | - | 18.7 |
| GPT-4o (partial API Hard) | - | 0.0 | - | 0.0 |
| Llama3.1-8B | 72.5 | 55.2 | 22.1 | 13.4 |
| Qwen2.5Math-7B | 30.4 | 15.7 | 9.3 | 0.0 |
| Qwen2.5Coder-7B | 79.1 | 60.6 | 24.9 | 15.6 |
| GPT-4o-mini | 81.0 | 61.7 | 23.8 | 14.4 |
| GPT-4o | 86.9 | 68.3 | 33.7 | 20.5 |
| Qwen2.5Coder-7B (SFT-Full) | 100 | 94.2 | 82.6 | 73.1 |
| GPT-4o (SFT-Full) | **100** | **98.8** | **94.3** | **89.6** |

Table 2: Comparison on single- and multi-domain tasks (arrows indicate higher is better).

## 4.2 DECOMPOSED MIP FORMULATION

Instead of formulating an M³P task as a single unified MIP problem, an alternative strategy is to decompose it into a sequence of single-domain subproblems, each solved as a smaller MIP in-

stance. For example, in the Footstep+UAV task, the LLM could predict a fixed waypoint by invoking `add_middle_point_constraints`, ensuring that the bipedal robot and the UAV meet at a common location. The LLM can then produce two separate MIP subproblems—one for the bipedal robot and one for the UAV—each guiding its respective agent to the designated waypoint.

Table 4 summarizes the results. SFT substantially improves code generation quality in this decomposed setting. Notably, ICL and SFT-Single achieve higher performance on multi-domain tasks than in the single MIP setting. This is likely because their in-context demonstrations and training data are structurally closer to single-task outputs, making decomposition more aligned with their learned priors.

| Method | Waypoint-Feasibility (%) |
|---|---|
| ICL | 24.3 ± 6.7 |
| SFT-Single | 47.0 ± 4.3 |
| SFT-Full | 80.5 ± 3.0 |

Table 3: LLM performance on generating feasible intermediate waypoints (avg ± std).

We further evaluate the feasibility of LLM-generated intermediate waypoints in Table 3. A feasible waypoint must lie within the environment bounding box and remain collision-free with respect to all obstacles. Results show that SFT markedly improves waypoint feasibility, rising from only 24% in ICL to over 80% in SFT-Full. This suggests that exposure to feasible samples during training provides a strong inductive bias for valid waypoint generation. However, despite the higher feasibility of intermediate waypoints, the overall success rate in the decomposed approach lags behind the single-MIP setting. This is because the end-to-end success requires the predicted waypoint to be feasible. The LLM's unstable performance in predicting waypoints introduces variance and occasional infeasible decompositions, which in turn reduce end-to-end success. These findings suggest a promising future direction: incorporating reinforcement learning for reasoning-based finetuning to encourage consistent intermediate waypoint generation.

| Task | ICL (avg ± std) | SFT-Single (avg ± std) | SFT-Full (avg ± std) |
|---|---|---|---|
| Finger Selection | 100.0±0.0 / 100.0±0.0 / 100.0±0.0 | 100.0±0.0 / 100.0±0.0 / 100.0±0.0 | 100.0±0.0 / 100.0±0.0 / 100.0±0.0 |
| Footstep Planning | 100.0±0.0 / 92.0±1.2 / 89.6±0.3 | 100.0±0.0 / 100.0±0.0 / 100.0±0.0 | 100.0±0.0 / 100.0±0.0 / 100.0±0.0 |
| Grasp | 100.0±0.0 / 88.0±3.0 / 77.4±1.2 | 100.0±0.0 / 100.0±0.0 / 100.0±0.0 | 100.0±0.0 / 100.0±0.0 / 100.0±0.0 |
| IK | 72.2±3.8 / 70.8±3.6 / 65.8±1.9 | 100.0±0.0 / 100.0±0.0 / 86.8±0.2 | 100.0±0.0 / 100.0±0.0 / 91.6±0.3 |
| UAV | 44.2±2.4 / 58.2±3.6 / 43.4±6.7 | 88.4±11.1 / 88.4±1.1 / 85.4±2.7 | 95.8±6.6 / 95.8±6.6 / 91.2±5.0 |
| Footstep + Footstep | 23.9±2.3 / 23.9±2.3 / 22.0±8.1 | 32.0±0.0 / 32.7±7.0 / 21.3±7.4 | 100.0±0.0 / 100.0±0.0 / 92.0±3.3 |
| Footstep + IK | 24.8±4.0 / 20.8±3.1 / 16.9±5.4 | 87.2±3.4 / 73.7±0.0 / 40.5±0.2 | 100.0±0.0 / 100.0±0.0 / 86.1±3.7 |
| Footstep + UAV | 21.5±1.9 / 15.6±1.1 / 10.4±2.7 | 55.2±0.0 / 16.8±0.0 / 7.5±2.2 | 100.0±0.0 / 100.0±0.0 / 66.7±1.8 |
| UAV + IK | 13.1±6.2 / 19.0±1.0 / 8.3±5.4 | 62.5±8.2 / 28.4±5.5 / 18.9±6.9 | 93.3±3.6 / 93.3±3.6 / 76.9±4.7 |
| UAV + UAV | 23.6±4.3 / 27.7±1.4 / 20.3±3.8 | 81.3±1.3 / 37.5±0.0 / 28.4±3.8 | 100.0±0.0 / 100.0±0.0 / 74.2±5.5 |
| Footstep + Grasp | 23.5±14.9 / 23.5±1.9 / 19.2±3.0 | 43.7±0.0 / 56.4±0.0 / 30.5±2.3 | 100.0±0.0 / 100.0±0.0 / 90.0±0.6 |
| UAV + Grasp | 5.2±0.6 / 14.9±1.4 / 2.0±0.0 | 34.5±0.0 / 37.5±0.0 / 13.0±4.9 | 100.0±0.0 / 100.0±0.0 / 82.8±0.0 |

Table 4: LLM has the option to decompose the single- and multi-domain tasks into multiple MIP subproblems. Comparison of Formulation-Correctness/Parameter-Correctness/Success-Rate(%) across ICL, SFT-Single, and SFT-Full.

## 5 CONCLUSION

We introduce LLM+MIP, a new paradigm for LLM-guided robot motion planning. Our approach builds on LLM+P (Liu et al., 2023), where an LLM is used to translate natural language descriptions of robot planning problems into formal representations that downstream search algorithms can solve, thereby significantly improving success rates. Unlike LLM+P, which is limited to symbolic action sequence reasoning, LLM+MIP translates problems into MIP instances, enabling joint reasoning over both discrete and continuous decision spaces. We demonstrate that LLM+MIP can solve various single- and multi-domain $M^3P$ tasks, a subset of TAMP. We also evaluate the LLM's ability as a translator for MIP formulations, showing that its success rate benefits substantially from SFT. Despite promising results, our method has several limitations that open avenues for future work. First, our dataset currently includes only 2D problem instances. Extending evaluation to 3D problems would be a valuable direction, though considerably more computationally expensive. Second, our method relies on existing discretization techniques. A promising future direction is to investigate whether LLMs can autonomously rediscover or even invent more efficient discretization strategies.

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

## A    TASK DESCRIPTION

We provide the detailed description of our five single-domain M³P tasks:

- Footstep Planning (Deits & Tedrake, 2014): The goal is for a bipedal robot to take a set of footsteps, with approximated footstep reachability constraints and obstacle-avoidance constraints, for its center of mass to reach a projected 2D position.
- Finger Selection (Hang et al., 2017): The goal is for a robot gripper to select contact points of each fingertip to maximize the 2D object grasping quality.
- 2D Inverse Kinematics (IK) (Dai et al., 2019): The goal is for a 2D articulated robot arm to reach a target configuration, as well as avoiding a set of obstacles.
- Grasp (Liu et al., 2020): This is the combination of finger selection and IK, where the robot gripper needs to select contact points as well as compute the pose for fingertips to reach each point.
- Collision-Free UAV Trajectory Planning (UAV) (Deits & Tedrake, 2015b): The goal is for a 2D UAV to fly to a certain target position without intersecting obstacles, while satisfying additional set of user-defined constraints. We support multiple types of constraints such as: "circle around a obstacle", "flying from the left of an obstacle"

We provide the detailed description of our seven multi-domain M³P tasks:

- Footstep+IK: The bipedal robot is equipped with a 2D articulated robot arm and needs to reach for distant goal position. Therefore it needs to first take several steps to walk close to the target and reach out for it.
- UAV+IK: Similar to Footstep+IK, but with a UAV equipped with a 2D articulated robot arm and needs to reach for distant goal position, while avoiding obstacles.
- Footstep+UAV: A bipedal robot and a UAV needs to meet at a known or unknown position, both avoiding obstacles.
- UAV+UAV: Similar to Footstep+UAV, but with two UAVs meeting at a known or unknown position.
- Footstep+Footstep: Similar to Footstep+UAV, but with two bipedal robots meeting at a known or unknown position.
- Footstep+Grasp: A bipedal robot needs to grasp a distant object, so it needs to first walk close to it, then select the grasping point, and finally reach out for it.
- UAV+Grasp: Similar to Footstep+Grasp, but with a UAV equipped with a gripper.

## B    LLM-CALLABLE API

In this section, we list the set of LLM-callable APIs for discretizing the general non-convex constraints into disjointly convex sets. We also implement a set of convenient functions for adding advanced constraints such as enforcing high-order continuity between pieces of UAV trajectory curves.

**`create_map(iris_num, obstacle_map)`**
> Generate IRIS convex regions with obstacles, defining the feasible environment.

**`add_iris_region_assignment_constraints(num_segments, pivots, footsteps, R_COLLISION)`**
> Constrain UAV trajectory segments, grasp collision points, robot arm pivots, or active footsteps to lie within safe IRIS regions. The parameter `R_COLLISION` enforces a clearance margin so that discretized points remain at least this distance away from region boundaries.

**`generate_side_and_vertical_obstacles(iris_regions, obstacle_map, key, style)`**
> Augment each obstacle with auxiliary boxes depending on its `style`. For

"left"/"right" a vertical slab and two horizontal boxes are added to form corridors; for "top"/"bottom" a horizontal slab and two vertical boxes enforce passage; for "circle" four bounding boxes approximate a circular exclusion zone.

**add_spatial_relation_constraints(text_lst, text_regions)**
Encode semantic navigation relations as region-assignment constraints. Directional tags ("left", "right", etc.) force the UAV to cross region sequences exactly once. "Circle" enforces cyclic transitions through four bounding regions, ensuring a complete detour.

**add_control_points_constraints(iris_regions, big_M)**
Link Bézier control points to selected IRIS regions via big-M constraints.

**add_continuity_constraints(num_segments, control_points)**
Enforce $C^0$, $C^1$, and $C^2$ continuity across Bézier segments. The parameter `control_points` are the decision variables defining the curve geometry.

**add_start_goal_constraints(start_pos, goal_pos)**
Fix the trajectory start and goal positions.

**add_base_constraints(BASE_LOCATION)**
Anchor the robot arm base to a fixed location.

**add_chain_rotation_constraints(LINK_LENGTHS, N_POLY)**
Encode multi-link kinematics using polynomial rotation approximations. `N_POLY` sets the polygon resolution for approximating trigonometric functions.

**add_end_constraints(END_LOCATION)**
Force the arm end effector to a target position.

**add_collision_discretization_constraints(pivots, NUM_INTERMEDIATE)**
Discretize each link into endpoints plus `NUM_INTERMEDIATE` samples for collision checking. Larger values yield finer safety resolution at higher cost.

**get_object_surface_samples(obj, delta, delta_n)**
Sample candidate contact points with spacing `delta`, and outward-shifted normals at offset `delta_n`.

**add_finger_selection_constraints(n_fingers, len_points)**
Select feasible contact points for `n_fingers` out of `len_points` candidates via binary variables.

**add_end_grasp_constraints(pivots, ee1, ee2)**
Constrain fingertip pivots to close consistently around the object. Ensure the first joint matches `ee1` and the last joint matches `ee2`.

**add_middle_point_constraints(middle_pivot, target_point)**
Anchor an intermediate pivot to a fixed target point.

**add_grasp_wrench_constraints(sample_points, sample_normals, selection_flag, friction_coef)**
Impose wrench feasibility and friction-cone stability given sample points, normals, contact flags, and friction coefficient.

**add_step_on_constraints(take, num_steps)**
Mark which footstep indices are active. The first two (initial stance feet) are forced active, while later steps are gated by binary `take` variables.

**add_monotonicity_constraints(take)**
Force `take` to be non-increasing, preventing reactivation of steps after termination.

**add_reachability_constraints(start_left, start_right, steps_taken, reachable_distance, foot)**
Constrain each new step to lie within a polygonal reachable region

relative to the previous stance. `start_left`/`start_right` fix the initial feet, `steps_taken` is the maximum planning horizon, `reachable_distance` sets the step-length bound, and `foot` specifies which side starts.

**`add_terminal_constraints(contact, take, term, steps_taken)`**
Select exactly one terminal step among the planning horizon. `contact` are continuous foot positions, `take` are binary transition variables, `term` are binary terminal selectors, and `steps_taken` sets the planning horizon.

**`share_point_feasibility_constraints(shared_points, subsystem_vars)`**
Tie multi-domain subsystems (UAV, IK, grasp, footsteps) to shared geometric points (e.g., UAV drop-off, grasp midpoint). `shared_points` are the common coordinates, `subsystem_vars` are the corresponding subsystem decision variables.

**`create_objective_and_solve(constraints, objective)`**
Assemble all constraints and define the optimization objective (step count, smoothness, feasibility, etc.). Call the solver (e.g., Gurobi) and return solution status, runtime, and variables.

## C    MIP FORMULATION

In this section, we provide the detailed formulation of each of our single-domain tasks.

### C.1    UAV

We formalize the UAV trajectory planning problem as a mixed-integer quadratic program (MIQP) (Deits & Tedrake, 2015b). Let $N$ denote the number of trajectory segments, $R$ the number of IRIS regions (Deits & Tedrake, 2015a), and $C \in \mathbb{R}^{4N \times 2}$ the cubic Bézier control points across all segments. Binary variables $H_{r,j} \in \{0, 1\}$ assign segment $j$ to region $r$. The MIQP adopts the following set of constraints:

**Region assignment:** Each segment must belong to exactly one region:

$$\sum_{r=1}^{R} H_{r,j} = 1, \quad \forall j = 1, \dots, N. \tag{1}$$

**Control point feasibility:** Let region $r$ be described by $A_r x \leq b_r$. For each segment $j$, control point $C_{j,k}$ ($k = 0, \dots, 3$) must lie in its assigned region, relaxed via the Big-M method:

$$A_r C_{j,k} \leq b_r + M(1 - H_{r,j}), \quad \forall r, j, k, \tag{2}$$

with $M$ being some large constant.

**Continuity:** Adjacent Bézier segments must connect smoothly in position, velocity, and acceleration. For $j = 1, \dots, N - 1$:

$$
\begin{aligned}
C_{j,3} &= C_{j+1,0}, \\
(C_{j,3} - C_{j,2}) &= (C_{j+1,1} - C_{j+1,0}), \\
(C_{j,3} - 2C_{j,2} + C_{j,1}) &= (C_{j+1,0} - 2C_{j+1,1} + C_{j+1,2}),
\end{aligned}
\tag{3}
$$

which can be added by API call `add_continuity_constraints`.

**Start/goal constraints:** The trajectory must start at $s$ and end at $g$:

$$C_{0,0} = s, \qquad C_{N-1,3} = g, \tag{4}$$

which can be added by API call `add_start_goal_constraints`.

**Objective:** We minimize the trajectory's integrated jerk, computed for each segment $j$ as $J_{j,d} = 6(C_{j,3,d} - 3C_{j,2,d} + 3C_{j,1,d} - C_{j,0,d})$ for dimension $d \in \{x, y\}$:

$$\arg\min_{C,H} \sum_{j=1}^{N} \sum_{d \in \{x,y\}} J_{j,d}^2. \tag{5}$$

Constraints Equation 1-Equation 4 together with objective Equation 5 define the UAV planning MIQP. This formulation tightly couples discrete region assignments with continuous Bézier control points, ensuring trajectories are collision-free, dynamically smooth, and solver-executable.

## C.2 Extension to Our Novel Spatial Constraints

We propose a novel type of constraint to enforce additional constraints on UAV, enabling more expressive trajectory descriptions. Specifically, we allow UAV to fly from the "left/right/top/bottom" of an obstacle, for a given number of times. We further allow UAV to circle around the obstacle, also for a given number of times. These additional requirements can be achieved by so-called crossing constraints. Suppose there are two IRIS regions indexed by $p$ and $q$, and we want to check whether UAV crosses the boundary of $p$ and $q$ at the $a$th trajectory segment. Such a check can be achieved by introducing the continuous indicator variable $W_{pq}^a$ and the following constraints:

$$W_{pq}^a \leq H_{p,a}, \quad W_{pq}^a \leq H_{q,a+1}, \quad W_{pq}^a \geq H_{p,a} + H_{q,a+1} - 1. \tag{6}$$

We can further ensure that only one crossing happens by the following constraints:

$$\sum_{a=1}^{N-1} W_{pq}^a + W_{qp}^a = 1. \tag{7}$$

Now, suppose we want the UAV to circle around an obstacle exactly once, then we can generate for IRIS regions surrounding the obstacle, as indexed by $b, r, t, l$ (short for bottom, right, top, and left). We then enforce the following constraints:

$$\sum_{a=1}^{N-1} W_{br}^a = \sum_{a=1}^{N-1} W_{rt}^a = \sum_{a=1}^{N-1} W_{tl}^a = \sum_{a=1}^{N-1} W_{lb}^a = 1. \tag{8}$$

Further suppose we want the UAV to fly from the left of an obstacle. Then we can generate a rectangular IRIS region on the left, as indexed by $l$. We then create two other IRIS regions, as indexed $l^t$ and $l_b$. $l^t$ intersects with $l$ from the top and $l_b$ intersects with $l$ from the bottom. With these three IRIS regions, we only need to enforce that the UAV fly $l^t$ to $l$ then to $l^b$, which is constrained by:

$$\sum_{a=1}^{N-1} W_{l^t l}^a = \sum_{a=1}^{N-1} W_{ll^b}^a = 1. \tag{9}$$

## C.3 IK

We present the simplified, 2D variant of Dai et al. (2019), modeling a planar manipulator with $L$ links of fixed lengths $\{\ell_1, \ldots, \ell_L\}$. Let $p_i \in \mathbb{R}^2$ denote the pivot of joint $i$, with base $p_0$ and end-effector $p_L$. Rotations are approximated via polygonal discretization with $N_{\text{poly}}$ facets. The MIP endows the following set of constraints.

**Base anchoring:** The base is fixed at a given location $s$:

$$p_0 = s. \tag{10}$$

**Rotation Discretization:** Each link's local to global transform uses a rotation matrix $R_i \in \mathbb{R}^{2 \times 2}$ constrained to lie in a polygonal approximation of $SO(2)$:

$$p_i = p_{i-1} + R_i \begin{bmatrix} \ell_i \\ 0 \end{bmatrix}, \quad i = 1, \ldots, L. \tag{11}$$

Each $R_i$ is constructed via piecewise linear relaxation of $SO(2)$ constructed by calling `add_chain_rotation_constraints`. This constraint is also used in footstep planning (Deits & Tedrake, 2014).

**End-effector constraint:** The final end-effector position must coincide with the goal $g$:

$$p_L = g. \tag{12}$$

**Collision discretization:** The link pose should be collision-free, which is formulated in a similar way as in UAV. Each link $[p_{i-1}, p_i]$ is discretized into $M$ intermediate points:

$$q_{i,k} = (1 - \tfrac{k}{M+1})p_{i-1} + \tfrac{k}{M+1}p_i, \quad k = 1, \ldots, M, \tag{13}$$

with all $\{p_i, q_{i,k}\}$ concatenated into collision point matrix $C_j$. Again we construct $R$ IRIS regions and use binary assignment variables $H_{r,j} \in \{0,1\}$ to map each collision point to exactly one region:

$$\sum_{r=1}^{R} H_{r,j} = 1, \quad \forall j. \tag{14}$$

Big-$M$ feasibility is enforced as:

$$A_r C_j \le (b_r - R_c) + M(1 - H_{r,j}) \quad \forall r, j, \tag{15}$$

where $R_c$ is some safety distance. Constraints Equation 10-Equation 15 define our MIQP. The formulation couples discrete rotation approximations, continuous kinematics, and region-based collision avoidance.

## C.4   GRASP

We present a simplified, 2D variant of Liu et al. (2020) for grasp planning. The grasp formulation extends the IK problem introduced in the previous section. All IK constraints (chain kinematics, collision discretization, IRIS region assignment, and optional spatial relation constraints) are inherited directly. We now introduce additional variables and constraints that model grasp contact selection and force-closure feasibility.

**Grasp selection (two fingers):**   We sample $N$ equidistant points on the surface of objects as candidate grasp points $\{p_i\}$, which is achieved by calling `get_object_surface_samples`. We further introduce binary matrices $F \in \{0,1\}^{2 \times N}$ and selection flags $s \in [0,1]^N$ select two distinct contact points from the $N$ surface samples, via the following constraints:

$$\sum_{i=1}^{N} F_{1i} = 1, \quad \sum_{i=1}^{N} F_{2i} = 1, \tag{16}$$

$$F_{1i} + F_{2i} \le 1, \ \ s_i = F_{1i} + F_{2i}, \quad i = 1, \dots, N. \tag{17}$$

The selected end-effector (contact) positions are

$$\text{ee}_1 = \sum_{i=1}^{N} F_{1i} \, p_i, \quad \text{ee}_2 = \sum_{i=1}^{N} F_{2i} \, p_i. \tag{18}$$

The above two variables replace the fixed start and goal constraints from the IK problem by anchoring the kinematic chain to two contact points on the object.

**Force-closure (wrench) constraints:**   For each sample $p_i$ with outward unit normal $n_i$ and tangent $t_i$, we define the two friction cone generators:

$$f_i^A = n_i + \mu \, t_i, \quad f_i^B = n_i - \mu \, t_i, \tag{19}$$

with corresponding planar wrenches $v_i^A = [\, f_i^A; \, (p_i \times f_i^A) \,] \in \mathbb{R}^3$ (and analogously $v_i^B$). Introducing nonnegatives $\alpha_{i,d}, \beta_{i,d}$ and slack $\gamma_d$ for each wrench direction $\hat{w}_d$, we impose

$$\sum_{i=1}^{N} (v_i^A \alpha_{i,d} + v_i^B \beta_{i,d}) = \gamma_d \, \hat{w}_d, \quad \forall d, \tag{20}$$

$$\alpha_{i,d} + \beta_{i,d} \le 1, \quad \alpha_{i,d} + \beta_{i,d} \le s_i, \quad \alpha_{i,d}, \beta_{i,d} \ge 0, \tag{21}$$

$$\gamma_d \ge r, \quad \forall d, \tag{22}$$

where $r$ is the common inscribed circle radius in wrench space.

**Objective:**   Our object is to maximize the inscribed circle radius, i.e. $\arg \max r$. Note that this inscribed circle radius can be approximated using discretization technique (Liu et al., 2020), leading to an MIQP, or the lower-bound relaxation (Dai et al., 2017), leading to an MISDP. We use the MIQP formulation since Gurobi does not solve MISDP. Put together, the grasp MIP augments the IK formulation with the above constraints, which jointly enforce valid finger selection and force-closure robustness.

### C.5 FINGER SELECTION

Finger selection can be viewed as a simplified version of the grasp problem (Section C.4), where we do not consider the reachability of fingers, and directly select $m$ contact points from $N$ sampled candidate grasp points to maximize the inscribed circle radius.

### C.6 FOOTSTEP

Footstep planning (Deits & Tedrake, 2014) builds upon the general spatial and region constraints introduced in the IK section, but specializes them for biped locomotion. In particular, footsteps must remain collision-free, lie in the IRIS regions, respect alternating left-right step ordering, and satisfy kinematic reachability constraints relative to the stance foot.

Let $N$ denote the maximum number of footsteps. Each step $i$ has a contact position $p_i = (x_i, y_i) \in \mathbb{R}^2$. As usual, we use a binary region assignment $H_{r,i} \in \{0, 1\}$ indicating whether step $i$ lies in region $r = 1, \ldots, R$. Further, a binary usage variable $u_i \in \{0, 1\}$ indicating whether step $i$ is active, and a binary terminal indicator $t_i \in \{0, 1\}$ specifies whether step $i$ is the terminal footstep. We use the following set of constraints.

**Region membership:** Each footstep must lie in exactly one IRIS region when active:

$$\sum_{r=1}^{R} H_{r,i} = u_i, \qquad A_r p_i \le b_r + M(1 - H_{r,i}), \quad \forall r, i. \tag{23}$$

Note that the trajectory between consecutive footsteps can have intersections with obstacles. This is because the robot can lift the foot to walk around the obstacles.

**Start and terminal constraints:** The first two contacts are fixed to the initial left/right feet:

$$p_0 = p_{\text{L}}^{\text{start}}, \quad p_1 = p_{\text{R}}^{\text{start}}, \tag{24}$$

and exactly one terminal step is chosen:

$$\sum_{i=0}^{N-1} t_i = 1, \qquad t_i \le u_i, \quad \forall i. \tag{25}$$

**Step activation monotonicity:** Once a step is inactive, no later step can be active:

$$u_i \ge u_{i+1}, \quad \forall i = 0, \ldots, N - 2. \tag{26}$$

This constraint can be added by calling `add_monotonicity_constraints`.

**Left-right ordering and separation:** Let steps alternate between left and right feet. We impose:

$$x_{i+1} - x_i \ge -M(1 - u_{i+1}) - \epsilon, \quad \text{if } i \text{ is left,} \tag{27}$$

$$x_i - x_{i+1} \ge -M(1 - u_{i+1}) - \epsilon, \quad \text{if } i \text{ is right,} \tag{28}$$

$$|x_{i+1} - x_i| \ge \Delta_{\min} - M(1 - u_{i+1}), \quad \forall i. \tag{29}$$

**Reachability:** From the stance foot $p_i$, the next contact $p_{i+1}$ must lie in the shifted reachable polygon $\mathcal{R}$ (approximated by $m$ halfspaces):

$$a_m^\top (p_{i+1} - p_i - s_i d) \le R + M(1 - u_{i+1}), \quad \forall m. \tag{30}$$

**Goal constraints:** Let $(x_T, y_T)$ denote the terminal foot location. Its deviation from the goal $(x_g, y_g)$ is measured by slack variables $g_x, g_y, u_x, u_y$:

$$g_x = x_T - x_g, \quad g_y = y_T - y_g, \tag{31}$$

$$u_x \ge \pm g_x, \quad u_y \ge \pm g_y. \tag{32}$$

**Objective:** We minimize a weighted sum of step usage, lane penalties, and goal deviation:

$$\min \; \alpha \sum_i \|p_{i+1} - p_i\|_1 + \beta \sum_i u_i + \gamma(u_x + u_y) + \sum_i \lambda_i, \tag{33}$$

where $\lambda_i$ denotes the penalty associated with step $i$ deviating from its preferred lane. Together, the above constraints define the footstep planning MIP: footsteps remain in IRIS regions, satisfy alternating gait and reachability, and reach the goal with a minimal number of steps.

### C.7 MULTI-DOMAIN TASKS

In multi-domain tasks, subsystems such as UAV, IK, and Grasp must coordinate at specific interface points—for example, the UAV endpoint aligning with the IK base, or the IK end-effector aligning with the Grasp target. These *shared points* enforce consistency across modalities while maintaining feasibility within each individual domain. While such coordination can be enforced using a standard MIP constraint, we instead use the specialized API call `share_point_feasibility_constraints` to more clearly convey the semantic intent of this coupling.

## D ROBOM³P DATASET EXAMPLE / LLM INPUT OUTPUT EXAMPLE

### D.1 SYSTEM PROMPT

**System Prompt**

```
You are a robotic optimization expert.  Your job is to read
a text instruction of a robot mission and translate it into
mixed-integer programming function-call trajectories.
```

**Output policy**:

- First, identify the task type from:
  UAV / Robot Arm IK / Finger Selection / Grasp /
  Footstep Planning / UAV + Robot Arm IK / UAV + Robot
  Arm IK Unknown Intermediate Point / UAV + Robot Arm
  IK Known Intermediate Point / Footstep + Robot Arm IK
  / Footstep + Robot Arm IK Unknown Intermediate Point
  / Footstep + Robot Arm IK Known Intermediate Point
  / UAV + Robot Arm IK + Grasp / Footstep + Robot Arm
  IK + Grasp / Footstep + UAV / UAV + UAV / Footstep +
  Footstep

- Produce the **COMPLETE** trajectory end-to-end (not just a
  single call).

- Respond with **ONLY** the trajectory in assistant.content
  (JSON mode).

- Preserve argument keys and list/dict shapes from the
  training distribution; do not rename fields or invent
  new ones.

- Separate consecutive calls by a single blank line only
  if you output a human-readable format; for JSON mode,
  output a single JSON value.

- Never include anything except the trajectory.

**Global Rules:**

- Select the task category from the mission description.

- For that category, you MUST emit the exact functions
  from below.

- You may ONLY change parameter values.  Keep parameter
  names/keys exactly as listed.

- For composition tasks (e.g., UAV + IK), emit the
  trajectory sequences for BOTH tasks as shown in
  in-context examples.

- For tasks with "Specify Unknown Shared Point," you MUST
  predict a suitable intermediate point that:

```
            – Does not collide with any obstacles
            – Is within reachable distance from the goal using
              the robot arm linkage length
            – Is positioned appropriately for the mission context

    Available Functions and Semantics:  Detailed in Section B.

    Composition Tasks:
            • The trajectory should be a combination of each
              individual task's trajectory sequence.
            • Use a shared variable (e.g., shared_point_1) to define
              intermediate points that connect tasks.
            • Use the shared variable as argument value for
              start_pos/end_pos or BASE_LOCATION/END_LOCATION where
              tasks connect.
            • Add share_point_feasibility_constraints(shared_variable)
              for each shared point.
            • Use create_objective_and_solve(combined_constraints) as
              the ONLY solver function at the end.

    Example composition structure:

    1. First task trajectory (e.g., UAV or Footstep)
    2. Second task trajectory (e.g., IK or Grasp)
    using shared_point as connection
    3. share_point_feasibility_constraints(point=shared_point)
    4. create_objective_and_solve(constraints=all_constraints)
```

## D.2 INVERSE KINEMATICS

**Text Command / LLM Input.**

With link lengths of 0.498m, 0.939m, and 0.725m, maneuver the robot arm from the base at $[-1.2603, -1.7615]$ to the end target at $[1.2218, -1.7720]$, ensuring to navigate through the right of `obst2`. Obstacles are provided in `obstacle_map.json` file.

obstacle_map.json: "obst1": "lb": [0.0, 0.0], "ub": [1.0, 1.0], "style": "none" ; "obst2": "lb": [-2.2, 0.0], "ub": [-1.8, 1.5], "style": "right" ; "obst3": "lb": [1.2, -0.5], "ub": [1.8, 0.5], "style": "none" .

**MIP Code Trajectory / LLM Output.**

Listing 1: Code for inverse kinematics tasks

```python
import cvxpy as cp

var_dict = {}
constraints = []

iris_regions, obstacles_list_plt, styles, obstacle_keys = create_map(
    iris_num=17,
    obstacle_map="obstacle_map.json"
)

iris_regions = generate_side_and_vertical_obstacles(
iris_regions=iris_regions,
    obstacle_map="obstacle_map.json", key="obst2", style="right"
)
```

```
base_constraints, p0, pivots = add_base_constraints(
    BASE_LOCATION=[-1.2603, -1.7615],
    var_dict=var_dict
)
var_dict["p_0"] = cp.Variable(2, name="p_0")
constraints.extend(base_constraints)

chain_constraints, Rs_list, pivots = add_chain_rotation_constraints(
    LINK_LENGTHS=[0.498, 0.939, 0.725],
    N_POLY=8,
    var_dict=var_dict,
    p0=p0,
    pivots=pivots
)
for idx in range(len(LINK_LENGTHS)):
    var_dict[f"p_{idx+1}"] = cp.Variable(2, name=f"p_{idx+1}")
constraints.extend(chain_constraints)

end_constraints = add_end_constraints(
    END_LOCATION=[1.2218, -1.7720],
    pivots=pivots
)
constraints.extend(end_constraints)

discretization_constraints, collision_points =
    add_collision_discretization_constraints(
        NUM_INTERMEDIATE=10,
        var_dict=var_dict,
        pivots=pivots
    )
for i in range(len(pivots) - 1):
    for k in range(1, NUM_INTERMEDIATE + 1):
        var_dict[f"q_link_{i}_pt_{k}"] = cp.Variable(
            2, name=f"q_link_{i}_pt_{k}"
        )
constraints.extend(discretization_constraints)

region_constraints, z_vars_list = add_iris_region_assignment_constraints(
    iris_env=iris_regions,
    R_COLLISION=0.02,
    var_dict=var_dict,
    collision_points=collision_points
)
for idx in range(len(collision_points)):
    var_dict[f"z_collision_{idx}"] = cp.Variable(
        len(iris_regions), boolean=True, name=f"z_collision_{idx}"
    )
    for r in range(len(iris_regions)):
        var_dict[f"y_collision_{idx}_{r}"] = cp.Variable(
            2, name=f"y_collision_{idx}_{r}"
        )
constraints.extend(region_constraints)

spatial_constraints = add_spatial_relation_constraints(
    styles="right",
    styles_related_regions=[[], [17,18,19], []],
    z_vars_list=z_vars_list,
    num_segments=len(pivots)
)
for k in range(len(pivots) - 1):
    var_dict[f"W_right_JI_{k}"] = cp.Variable(
        boolean=True, name=f"W_right_JI_{k}"
    )
    var_dict[f"W_right_IJ_{k}"] = cp.Variable(
```

```
          boolean=True, name=f"W_right_IJ_{k}"
      )
constraints.extend(spatial_constraints)

status, ik_result = create_objective_and_solve(
    constraints=constraints,
    var_dict=var_dict,
    objective='feasiable',
    pivots=pivots,
    Rs_list=Rs_list
)
```

### D.3 UAV TRAJECTORY GENERATION

**Text Command / LLM Input.**

Depart from the highland outpost and make your way to the meadow sanctuary, ensuring the red brick warehouse remains on your right, the ancient oak grove stays on your left, and you fly above the sunken courtyard from the bottom. Obstacles are provided in `obstacle_map.json`.

obstacle_map.json: "red brick warehouse": "lb": [0.0, 0.0], "ub": [1.0, 1.0], "style": "right" ; "ancient oak grove": "lb": [-9.2, 0.0], "ub": [-8.8, 1.5], "style": "left" ; "sunken courtyard": "lb": [-6.9, -0.5], "ub": [-0.8, 0.5], "style": "above" ; "farm": "lb": [2.9, 2.5], "ub": [3.8, 3.5], "style": "none"

**MIP Code Trajectory / LLM Output.**

Listing 2: UAV trajectory code

```
import cvxpy as cp

var_dict = {}
constraints = []
NUM_SEGMENTS = 24
BEZIER_SEGMENTS = 4

iris_regions, obstacles_list_plt, styles, obstacle_keys = create_map(
  iris_num=36,
  obstacle_map="obstacle_map.json"
)

iris_regions = generate_side_and_vertical_obstacles(
  iris_regions=iris_regions, key='red_brick_warehouse', style='right'
)
iris_regions = generate_side_and_vertical_obstacles(
  iris_regions=iris_regions, key='ancient_oak_grove', style='left'
)
iris_regions = generate_side_and_vertical_obstacles(
  iris_regions=iris_regions, key='sunken_courtyard', style='bottom'
)

var_dict["H"] = cp.Variable(
(len(iris_regions), NUM_SEGMENTS), boolean=True, name="H")
collision_points = [var_dict["C"][i,:]
for i in range(NUM_SEGMENTS*BEZIER_SEGMENTS)]

region_constraints, z_vars_list = add_iris_region_assignment_constraints(
  iris_env=iris_regions,
  R_COLLISION=0.02,
  var_dict=var_dict,
  collision_points=collision_points
)
constraints.extend(region_constraints)
```

```
1242
1243  var_dict["W_right"] = [
1244      cp.Variable(boolean=True, name=f"W_right_{k}")
1245      for k in range(NUM_SEGMENTS-1)
1246  ]
1247  var_dict["W_left"] = [
1248      cp.Variable(boolean=True, name=f"W_left_{k}")
1249      for k in range(NUM_SEGMENTS-1)
1250  ]
1251  var_dict["W_bottom"] = [
1252      cp.Variable(boolean=True, name=f"W_bottom_{k}")
1253      for k in range(NUM_SEGMENTS-1)
1254  ]
1255  spatial_constraints = add_spatial_relation_constraints(
1256    var_dict["H"],
1257    text_lst=['right', 'left', 'bottom'],
1258    text_regions=[[36,37,38], [39,40,41], [42,43,44]],
1259    num_segments=NUM_SEGMENTS
1260  )
1261  constraints.extend(spatial_constraints)
1262
1263  var_dict["C"] = cp.Variable((NUM_SEGMENTS*BEZIER_SEGMENTS, 2), name="C")
1264  control_constraints = add_control_points_constraints(
1265    var_dict["C"], var_dict["H"],
1266    iris_regions=iris_regions, num_segments=BEZIER_SEGMENTS,
1267    big_M=10.0
1268  )
1269  constraints.extend(control_constraints)
1270
1271  continuity_constraints = add_continuity_constraints(
1272    var_dict["C"], num_segments=BEZIER_SEGMENTS
1273  )
1274  constraints.extend(continuity_constraints)
1275
1276  start_goal_constraints = add_start_goal_constraints(
1277    var_dict["C"],
1278    start_pos=obstacle_map['highland_outpost'],
1279    goal_pos=obstacle_map['meadow_sanctuary'],
1280    num_segments=BEZIER_SEGMENTS
1281  )
1282  constraints.extend(start_goal_constraints)
1283
1284  status, cost, H_val, C_val = create_objective_and_solve(
1285    constraints=constraints,
1286    var_dict=var_dict,
1287    objective='uav_jerk',
1288    C=var_dict["C"]
1289  )
```

## D.4 UAV+IK

**Text Command / LLM Input.**

Environment with obstacles (obst1 from [-2.0, -1.0] to [-0.5, 0.0], obst2 from [1.0, 0.0] to [2.0, 1.0], obst3 from [0.0, -2.0] to [2.0, -1.0]). Because the goal is too far to directly perform arm inverse kinematics with the 1.5 m arm (links [0.5, 0.5, 0.5]), the UAV first flies from $[-1.6, -1.6]$ through safe regions to an intermediate shared point, where the robot arm is then deployed to reach $[1.4, 1.4]$.

**MIP Code Trajectory / LLM Output.**

Listing 3: Unified UAV+IK trajectory code

```
# --- UAV trajectory setup ---
# (same as UAV case, up to start/goal constraints, omit here for clarity)
....

start_goal_constraints = add_start_goal_constraints(
    var_dict["C"],
    start_pos=[-1.6, -1.6],
    goal_pos=shared_uav_ik_base,
    num_segments=BEZIER_SEGMENTS
)

# --- Shared point linking UAV and IK ---
var_dict["shared_point"] = cp.Variable(2, name="shared_uav_ik_base")

# Shared point IRIS membership decision variables
var_dict["sp_z"] = cp.Variable(
len(iris_regions), boolean=True, name="z_shared_point")
var_dict["sp_y"] = [
    cp.Variable(2, name=f"y_shared_point_{r}")
    for r in range(len(iris_regions))
]

# --- IK chain setup ---
# (same as IK case, omit here for clarity)
....

constraints.extend(add_base_constraints(
    BASE_LOCATION=shared_uav_ik_base,
    var_dict=var_dict
))

# Explicit feasibility constraints for shared point
constraints.extend(share_point_feasibility_constraints(
    point=shared_uav_ik_base
))

# --- Solve unified problem ---
status, result = create_objective_and_solve(
    constraints=constraints,
    var_dict=var_dict,
    objective="uav_jerk",
    pivots=pivots,
    Rs_list=Rs_list
)
```

## D.5 FINGER SELECTION

**Text Command / LLM Input.**

To stably grasp an object using 3 fingers, select three grasp points that form a triangle around the object's center of mass, ensuring each point is on a stable surface and allows for equal distribution of force. Grasp object in obstacle_map.json: "obj": "triangle": "vertices": [ [0.0, 0.0], [1.0, 0.0], [0.5, 0.8] ] , "obj:edge": "edge1": "vertices": [ [0.0, 0.0], [1.0, 0.0] ] , "edge2": "vertices": [ [1.0, 0.0], [0.5, 0.8] ] , "edge3": "vertices": [ [0.5, 0.8], [0.0, 0.0] ]

**MIP Code Trajectory / LLM Output.**

Listing 4: Finger selection trajectory code

```
import cvxpy as cp

var_dict = {}
```

```
constraints = []

sampled_points, sampled_normals = get_object_surface_samples(
  obj=obj,
  delta=0.1,
  delta_n=0.02
)

var_dict["finger_selection"] = cp.Variable(
(3, len(sampled_points)), boolean=True, name="finger_selection")

var_dict["selection_flags"] = cp.Variable(
len(sampled_points), nonneg=True, name="selection_flags")

finger_constraints, finger_selection_var, selection_flags =
add_finger_selection_constraints(
  n_fingers=3,
  len_points=len(sampled_points)
)
constraints.extend(finger_constraints)

var_dict["r"] =
cp.Variable(name="radius_of_inscribed_circle", nonneg=True)
directions = sample_sphere(subdivisions=0)

wrench_constraints = add_wrench_constraints(
  points=sampled_points,
  normals=sampled_normals,
  selection_flags=var_dict["selection_flags"],
  r=var_dict["r"],
  directions=directions,
  friction_coef=1.0
)
constraints.extend(wrench_constraints)

status, finger_selection_val, r_opt = create_objective_and_solve(
  constraints=constraints,
  var_dict=var_dict,
  objective='finger',
  r=var_dict["r"],
  finger_selection_var=var_dict["finger_selection"]
)
```

## E  MORE RELATED WORK

**TAMP and M³P:** At its core, TAMP integrates two complementary layers of reasoning: a symbolic planner that selects and orders abstract actions, and a downstream motion planner that instantiates these actions as feasible continuous motions in the robot's configuration space. Recent years have seen notable advances in both the generality of task planning methods (Helmert, 2006; Piotrowski et al., 2024) and their efficiency and robustness (Dantam et al., 2018; Thomason et al., 2022). Despite this progress, the coupling between the high-level symbolic planner and the low-level motion planner remains a key design challenge. The two components are typically connected through a relatively loose interface, where symbolic reasoning provides candidate actions or parameters, and the motion planner either confirms feasibility or signals failure, prompting the symbolic layer to refine its proposals (Erdem et al., 2011; Akbari et al., 2015). This geometric view of M³P naturally lends itself to more integrated and efficient search strategies (Kingston et al., 2020; Beyer et al., 2021; Kingston & Kavraki, 2022), since planning across modes can be cast as finding feasible transitions between manifolds rather than treating each mode in isolation.

**LLM-as-Planner and LLM-with-Planner:** While intuitive and flexible, the LLM-as-Planner approach is fundamentally constrained by the limited long-horizon reasoning ability of current LLMs. Building on this idea, subsequent works have continued to expand the LLM-with-Planner paradigm.

For instance, Silver et al. (2024) investigated whether LLMs can generalize across multiple planning domains specified in PDDL. More recently, Agarwal et al. (2025) introduced structured memory into the LLM+Planner pipeline, allowing it to track dynamic world states more effectively.

## F RESULT VISUALIZATION

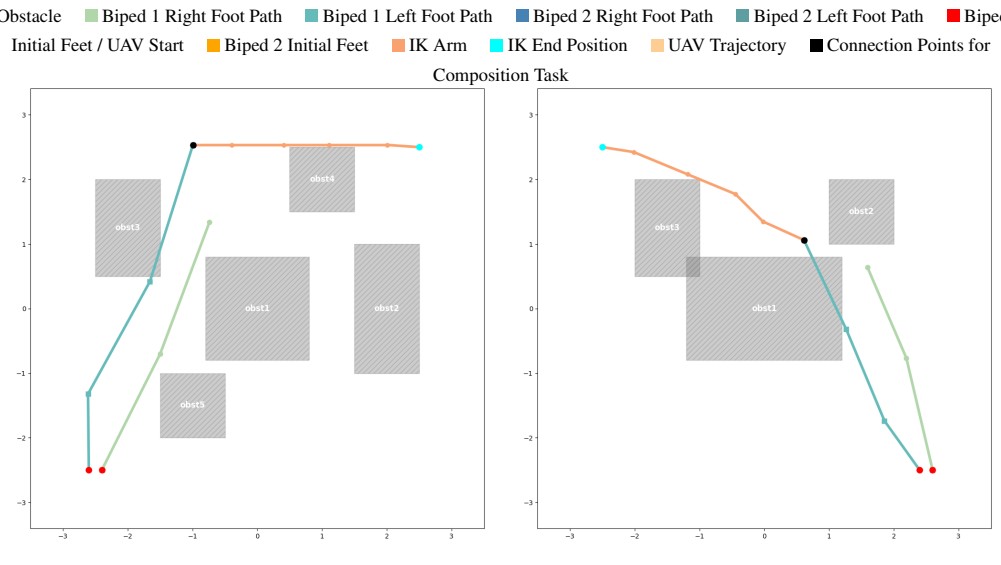

(a) A biped robot at [-2.6,-2.5] & [-2.4,-2.5] (left&right foot) with an arm configuration [0.2,0.4,0.4,0.4,0.2] and its arm needs to pass through bottom of obst6 and touch a target at [2.5,2.5]. Since the arm is too short for direct inverse kinematics, plan the trajectory so the robot walks to a feasible position to reach it. Robot Arm must pass obst4 top.

(b) A biped robot at [2.4,-2.5] & [2.6,-2.5] (left&right foot) with arm [0.5,0.9,0.9,0.9,0.5] must reach a target at [-2.5,2.5]. Because the arm length is insufficient, plan the trajectory so the robot relocates to a feasible place to complete the task.

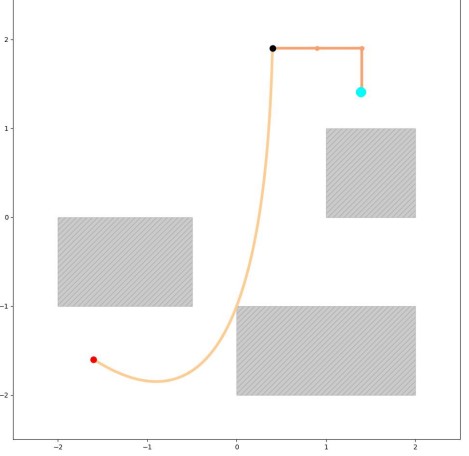

(c) Starting at [-1.5,-1.5] with arm [0.5,0.5,0.5], the UAV cannot directly reach the target at [1.5,1.5]. Plan the trajectory such that the UAV relocates to a position where inverse kinematics becomes feasible.

(d) A UAV at [2,2] with arm [0.6,0.6,0.5] must fly from the left of obstacle D touch a target at [1,0]. Because the arm is insufficient, plan the trajectory so the UAV moves into a feasible place to reach the point.

Figure 7: Additional Visual Result 1.

## G USAGE OF LLMS

We acknowledge the use of large language models (LLMs) as assistive tools in this research. LLMs are used during paper writing, for improving grammar and wording. All outputs from these models

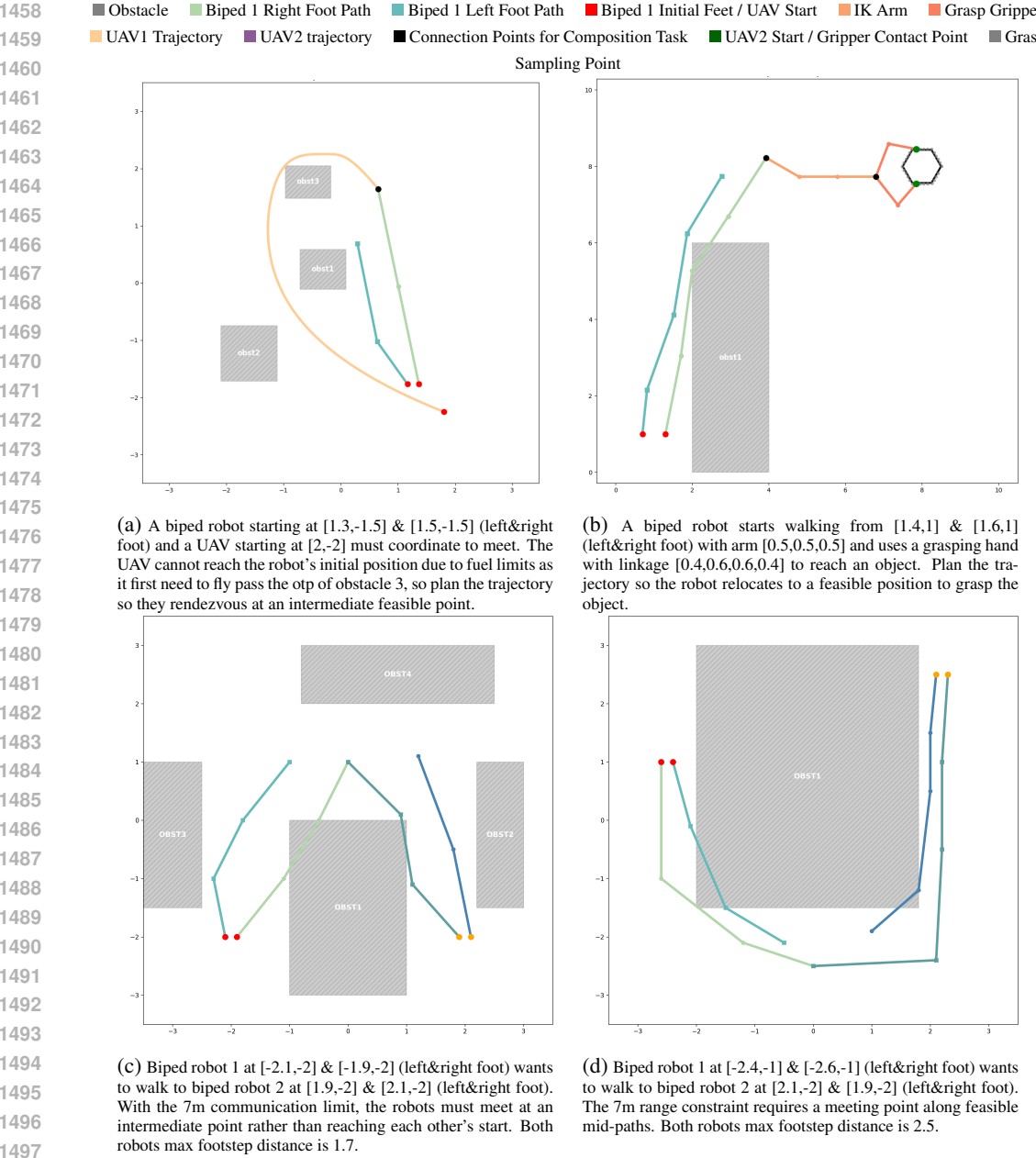

(a) A biped robot starting at [1.3,-1.5] & [1.5,-1.5] (left&right foot) and a UAV starting at [2,-2] must coordinate to meet. The UAV cannot reach the robot's initial position due to fuel limits as it first need to fly pass the otp of obstacle 3, so plan the trajectory so they rendezvous at an intermediate feasible point.

(b) A biped robot starts walking from [1.4,1] & [1.6,1] (left&right foot) with arm [0.5,0.5,0.5] and uses a grasping hand with linkage [0.4,0.6,0.6,0.4] to reach an object. Plan the trajectory so the robot relocates to a feasible position to grasp the object.

(c) Biped robot 1 at [-2.1,-2] & [-1.9,-2] (left&right foot) wants to walk to biped robot 2 at [1.9,-2] & [2.1,-2] (left&right foot). With the 7m communication limit, the robots must meet at an intermediate point rather than reaching each other's start. Both robots max footstep distance is 1.7.

(d) Biped robot 1 at [-2.4,-1] & [-2.6,-1] (left&right foot) wants to walk to biped robot 2 at [2.1,-2] & [1.9,-2] (left&right foot). The 7m range constraint requires a meeting point along feasible mid-paths. Both robots max footstep distance is 2.5.

Figure 8: Additional Visual Result 2.

were meticulously reviewed, revised, and verified by the authors, who retain full responsibility for all content presented in this paper.

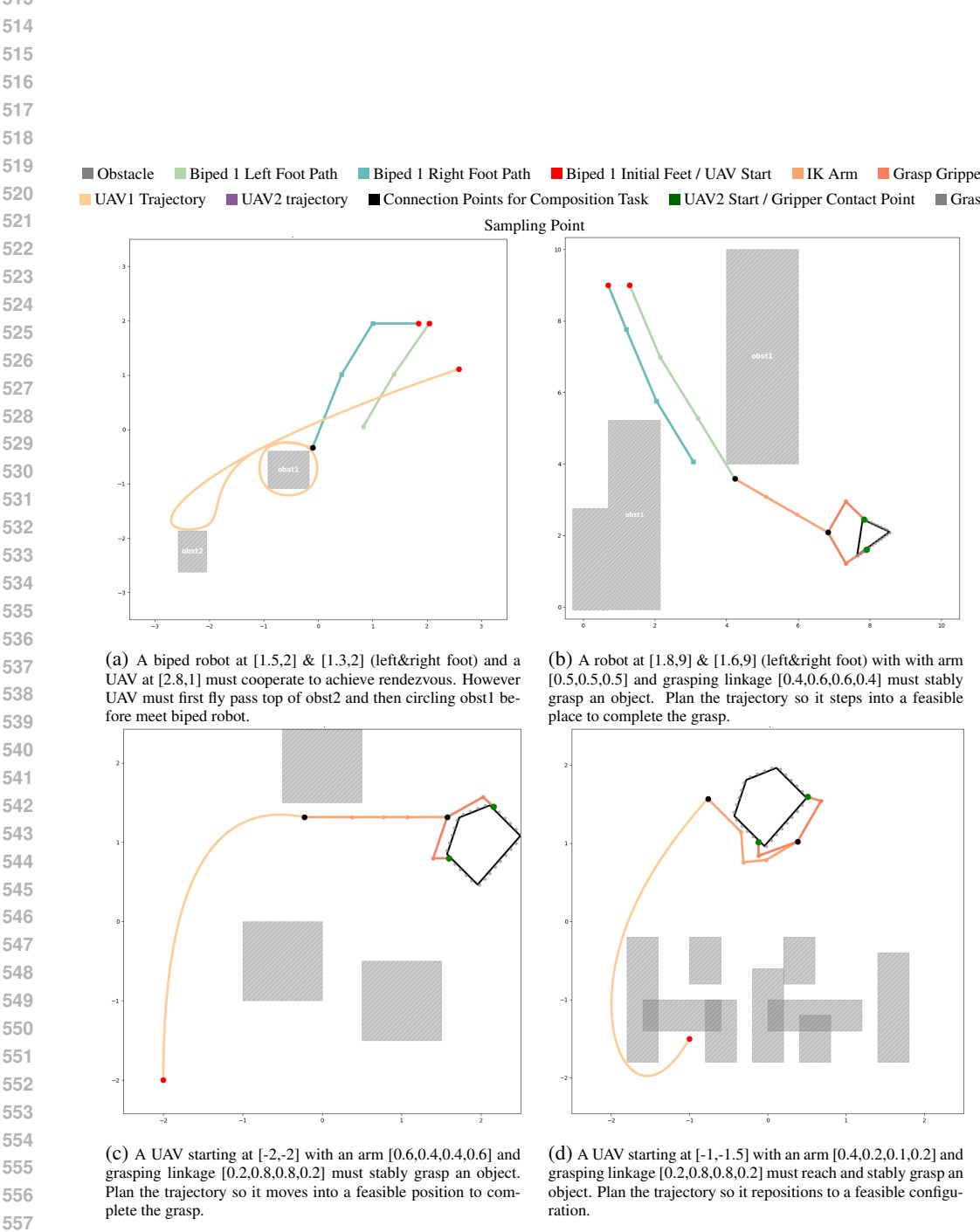

(a) A biped robot at [1.5,2] & [1.3,2] (left&right foot) and a UAV at [2.8,1] must cooperate to achieve rendezvous. However UAV must first fly pass top of obst2 and then circling obst1 before meet biped robot.

(b) A robot at [1.8,9] & [1.6,9] (left&right foot) with with arm [0.5,0.5,0.5] and grasping linkage [0.4,0.6,0.6,0.4] must stably grasp an object. Plan the trajectory so it steps into a feasible place to complete the grasp.

(c) A UAV starting at [-2,-2] with an arm [0.6,0.4,0.4,0.6] and grasping linkage [0.2,0.8,0.8,0.2] must stably grasp an object. Plan the trajectory so it moves into a feasible position to complete the grasp.

(d) A UAV starting at [-1,-1.5] with an arm [0.4,0.2,0.1,0.2] and grasping linkage [0.2,0.8,0.8,0.2] must reach and stably grasp an object. Plan the trajectory so it repositions to a feasible configuration.

Figure 9: Additional Visual Result 3.

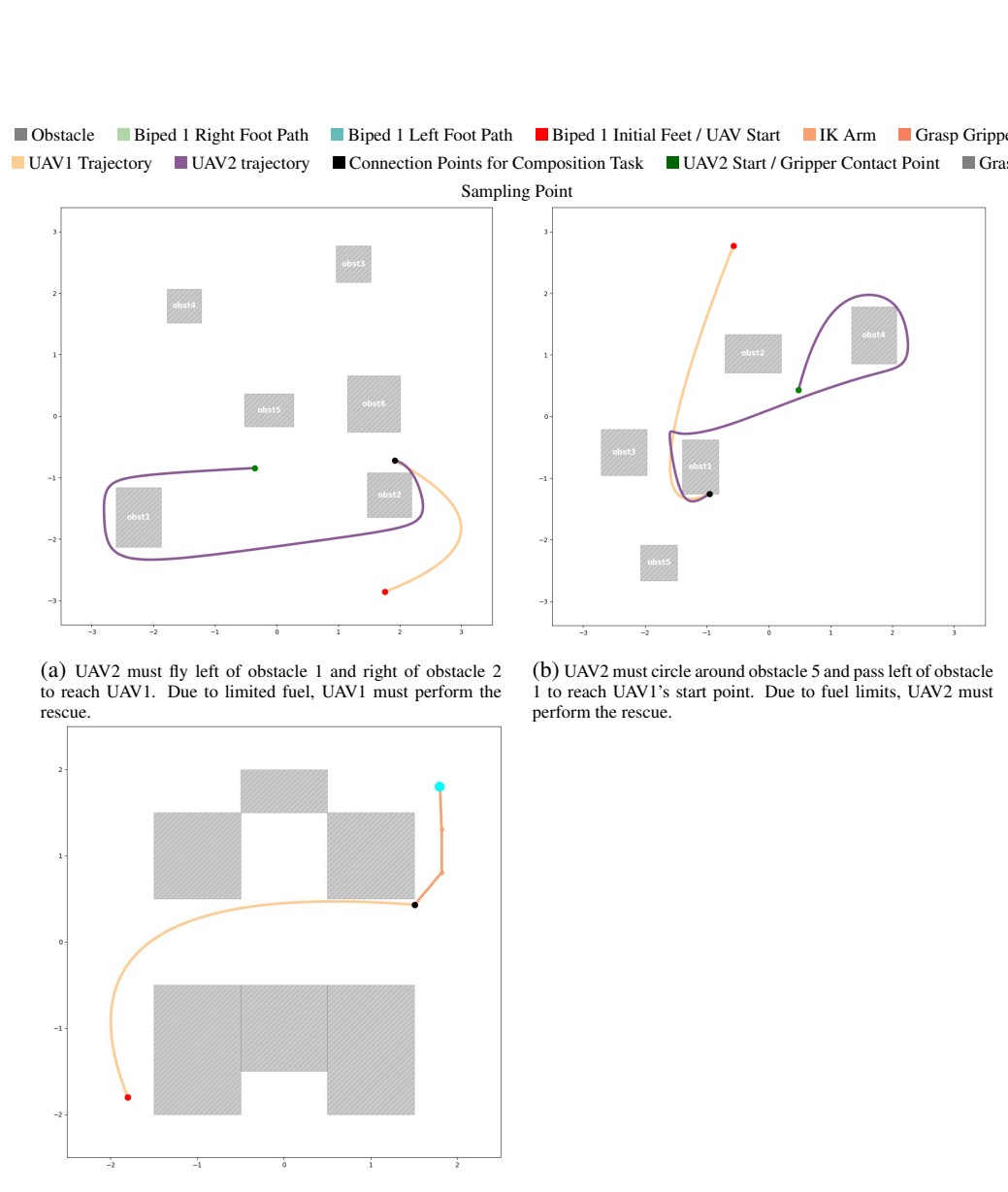

(a) UAV2 must fly left of obstacle 1 and right of obstacle 2 to reach UAV1. Due to limited fuel, UAV1 must perform the rescue.

(b) UAV2 must circle around obstacle 5 and pass left of obstacle 1 to reach UAV1's start point. Due to fuel limits, UAV2 must perform the rescue.

(c) A UAV starts at [-1.8,-1.8] with an arm configuration [0.5,0.5,0.5] and needs to reach a target at [1.8,1.9]. Since the arm is too short for direct inverse kinematics, plan the trajectory so the UAV moves to a feasible position to complete the task.

Figure 10: Additional Visual Result 4.

