# OpenReview forum: "Large Language Model-guided Multi-modal Motion Planning via Mixed Integer Program"
_ICLR.cc/2026/Conference — ICLR 2026 Conference Withdrawn Submission_

### Official Review · Reviewer_LZS8 · 2025-10-22

**Soundness:** 3
**Presentation:** 3
**Contribution:** 2
**Rating:** 2
**Confidence:** 4

**Summary:**

A common approach for robot motion planning involves mixed integer programming (MIP), eg for footstep planning in humanoids. Such approaches rely on the convexity of the problem formulation to be efficient. In practise, this can be a limiting factor, as the problem formulation is not trivial and commonly requires approximations to guarantee convexity. The present paper proposes to address this limitation by leveraging an LLM to propose convex problem approximation for MIP. The method, RoboM3P, uses an LLM to make API calls to write a python script that defines an optimization problem which is then solved using the Gurobi solver. In addition to calling a method to construct a convex decomposition of the configuration space, the method can impose additional constraints, eg, on the order of convex regions that shall be visited via calls of the API. The evaluation is carried out on 2D tasks, showing that finetuned models perform the best in this setup.

**Strengths:**

- The finetuning analysis is interesting. To the best of my knowledge, this is at least one of the first works that goes beyond using off-the-shelf LLMs for code generation in robotics.
- The idea of the paper is sound, simple and intuitive.

**Weaknesses:**

### Method
- The paper investigates the utility of LLMs for writing MIPs in the context of robot motion planning. I think this is an interesting problem and I think the idea of using an LLM to design the optimization problems is promising. I think given that prior work has already proposed this idea, the paper lacks some additional contribution to be accepted as it is. This method in itself is fine, but I think the insight alone that LLMs can write such MIPs is not convincing, as this appears to be relatively clear in light of the works of the last 1-2 years [3-5]. I think if the paper would expand their evaluation / method to investigate more what should be considered when following this approach would be very interesting. For instance, how do we need to prompt the models to achieve good performance, how should numerical values be treated given that LLMs still tend to struggle with floats, are the MIPs in robotics any different to general MIPs designed by LLMs?
- Including every type of task separately in the system prompt does not seem very scalable to me, which is always a promise that is associated with LLMs.

### Wording:
- "Recent research has explored the use of Mixed-Integer Programming (MIP) to reformulate global search problems in robotic applications". Using MIP for humanoid motion planning is not a recent advancement, eg, for Atlas this has been used for a decade [1]. You also cite some of this work in the introduction.
- "Given the limitations of traditional TAMP, especially the weak coupling between task and motion planning, researchers have increasingly explored unified formulations and integrated solver" This sentence is very abstract to me. What is the weak coupling? Also the second part of the sentence makes it sound like the work that you go on to cite is recent, but PDDL-based approaches are standard by now as you point out yourselves by citing 20 year old references.
- "As a result, TAMP systems must interface symbolic planners with separate motion planners (Garrett et al., 2021), often through heuristics or manually engineered bridging layers" This is not true. There are notable approaches such as LGP [2] (a variant thereof you cite yourselves in Toussaint et al, 2018) that formulate the problem as a single big optimization problem that does not rely on separate motion planners. I suggest you reformulate this sentence.

### Evaluation
- The evaluation focuses on simple problems in 2D. Given that prior work in this research direction solves robotics problems in >=7D in sim and real world, I think the presented experiments are relatively weak examinations of the system performance and lack in comparison to [3-5].
- I think you should compare to an extra modality which allocates compute for inference instead of finetuning since this has become a predominant approach over the last 2 years. Also in robotics this has been shown to work well and can replace finetuning [3].
- I dont really know what the rules on this are at ICLR, but the experiments only evaluate on models (GPT-4o) for which its no surprise that they are rather poor. I would think that more recent model iterations will be much better at coding, given current leaderboards on competitions and benchmarks [6]. But I also know that those are usually proprietary and expensive, so I wouldnt reject the paper just because of the model choice here, but nevertheless, evaluation would be stronger if some model like GPT-4.5

#### Clarity
- In your introduction and related work you make it sound like the idea of letting an LLM write motion planning problems is a key novelty paper of this paper. However, there has been quite a bit of work on this recently [3-5]. None of these works proposes to write MIPs in the way that you do, so I think you present some novelty, but I think it mid like the idea of letting an LLM write motion planning problems is a key novelty paper of this paper. However, there has been quite a bit of work on this recently [3-5]. None of these works proposes to write MIPs in the way that you do, so I think you present some novelty, but I tght be worth to discuss some approaches that use LLMs to design motion / M3P / TAMP problem formulations that will then be solved by an optimizer. Also you should change your wording in the introduction accordingly.
- "Detailed examples of these data pairs, are provided in the Section D.2, Section D.3, Section D.5. T" it would be good if you could include at least one abbreviated example in the main paper
- The legend in Fig 6 is overloaded and difficult to understand. You should try to reduce the number of colors and labels somehow.

#### Minor
- "Given recent advances in the mathematical reasoning capabilities of LLM" should be "LLMs" in plural here.
- Somehow the gaps after the table captions are sometimes quite small which make the text pretty hard to read, eg, Table 1. Can you somehow increase this?

---
### Sources
[1] Kuindersma, Scott, et al. "Optimization-based locomotion planning, estimation, and control design for the atlas humanoid robot." _Autonomous robots_ 40.3 (2016): 429-455.

[2] Toussaint, Marc. "Logic-Geometric Programming: An Optimization-Based Approach to Combined Task and Motion Planning." _IJCAI_. 2015.

[3] Curtis, Aidan, et al. "Trust the PRoC3S: Solving Long-Horizon Robotics Problems with LLMs and Constraint Satisfaction." _Conference on Robot Learning_. PMLR, 2025.

[4] Shcherba, Denis, et al. "Meta-Optimization and Program Search using Language Models for Task and Motion Planning." _arXiv preprint arXiv:2505.03725_ (2025).

[5] Mendez-Mendez, Jorge. "A Systematic Study of Large Language Models for Task and Motion Planning With PDDLStream." _arXiv preprint arXiv:2510.00182_ (2025).
[6] https://lmarena.ai/

**Questions:**

- "The MIP code is expressed as a structured sequence of function-like API calls, for example: xxx_constraint(arg1=val1)" what is xxx here?
- How do you actually generate the dataset? You discuss the composition in detail but I fail to understand where you retrieve these instances from. This is a crucial part of the paper, so it should become clear.
- How can Formulation-Correctness be evaluated? Is there always a unique correct set?
- What about syntactic correctness? Do you have any issues of instructuion following with the LLMs, eg, using API calls that dont exist? At least the main paper does not discuss that which surprises me.
- How does the number of in context examples influence the outcome of ICL?
- Are the 2d domain plots in Fig 6 inputs to the model as well or only the text?
- Are the trajectories that intersect the grey boxes in Fig 6 infeasible?
- Do you run the open models locally or via some API?

---

### Official Review · Reviewer_4T7n · 2025-10-31

**Soundness:** 2
**Presentation:** 2
**Contribution:** 2
**Rating:** 2
**Confidence:** 4

**Summary:**

The paper presents LLM+MIP, a framework that utilizes a large language model to translate natural-language multi-modal motion planning (M^{3}P) problems into mixed-integer programs (MIPs) via a callable CVXPY-Gurobi interface with discretization APIs. The core idea is that LLMs can automatically construct optimization-based motion plans by composing primitive geometric and kinematic constraints. The framework achieves notable success rates of 98.8% for single-domain tasks and 89.6% for multi-domain tasks, using supervised fine-tuning on the RoboM^{3}P dataset. While the motivation of the work is compelling and the engineering effort substantial, the work overstates the significance of "LLM-guided" planning and falls short of control-theoretic rigor. The absence of closed-loop analysis, robustness evaluation, or theoretical modeling weakens the scientific contribution, reducing the work to a text-to-MIP code generation study rather than an advancement in control or motion planning.

**Strengths:**

1. The authors identify an interesting research realm between symbolic planners (LLM+P) and continuous optimization-based planners, framing MIP as a bridge between discrete and continuous planning. The introduction of the RoboM^{3}P dataset, which includes structured MIP code, obstacle maps, and task descriptions, provides a valuable asset for studying mathematical reasoning in LLMs.

2. The discretization APIs (e.g., IRIS for collision avoidance, spline and reachability constraints, piecewise-linear IK approximations) are implemented cleanly and systematically exposed to the LLM. The code listings in the appendices and the dataset statistics, as presented in Figures 2-4, are well-documented and reproducible.

3. The experiments cover multiple M^{3}P domains, such as locomotion, UAV, IK, grasp, and multi-domain compositions. The success metrics, including Formulation Correctness, Parameter Correctness, and Feasibility, show SFT consistently outperforming ICL baselines (Table 4). The dataset analysis (e.g., average of 15.1 function calls and 45.4 arguments per trajectory) demonstrates the nontrivial combinatorial complexity the LLM must handle.

**Weaknesses:**

1. The proposed framework focuses entirely on offline formulation synthesis with no consideration of feedback, dynamics, or robustness. The generated MIPs are solved in an open-loop setting and not validated in receding-horizon or closed-loop settings. This actually disconnects from control principles undermines claims of "end-to-end" or "reasoning-based" planning. Classical mixed-integer planning and control frameworks, such as [1, 2, 3], which have already integrated optimization and feedback guarantees within mixed-integer formulations, highlight the theoretical gap in this area.

2. The system's success depends heavily on handcrafted discretization APIs and domain-specific libraries. The LLM merely orchestrates these components, performing templated code synthesis rather than reasoning or optimization. In essence, the "intelligence" resides in the library, not the model; this is a point that is unacknowledged in the discussion.

3. The evaluation metrics (Formulation Correctness and Feasibility) are proxies for syntactic and geometric consistency but reveal nothing about control performance, which is a crucial aspect in planning and control. Indeed, there are no measures of trajectory stability, robustness under disturbances, constraint violation rates, or real-time feasibility. Without these, it is impossible to assess whether LLM+MIP provides meaningful advantages over established control formulations such as contact-implicit trajectory optimization or task-and-motion planners with provable consistency, as in [4, 5, 6].

4. The method chains multiple approximations without quantifying optimality loss or feasibility margins. The system does not incorporate raw sensing, lacks dynamics and environmental uncertainty, and the LLM is unaware of low-level control; it generates code that a classical solver executes offline. The contributions are primarily focused on code synthesis for modeling, as well as dataset curation, rather than an end-to-end robotic approach. Additionally, there is no runtime or scaling analysis to show how discretization density or MIP size affects solvability.

5. The reported generalization is limited to API-driven 2D toy tasks. The extension to the spatial domain or high-dimensional planning is crucially needed. Throughout the paper, there is no 3D perception, noise evaluation, or comparison against modern TAMP frameworks or model-based planners. The claim that LLM+MIP unifies discrete and continuous planning is conceptually correct but empirically shallow, as it does not outperform existing hybrid optimization-based controllers on control or efficiency grounds.

6. Some aspects are suggested and should be improved in this version, including: (1) establishing robust feasibility or approximation bounds for discretization and big-M relaxations, (2) integration of a receding-horizon or MPC wrapper to test closed-loop robustness, (3) evaluations against control baselines, those of non-LLM template generator, with provable guarantees, and (4) quantifications of runtime, scalability, and approximations.

Overall, LLM+MIP is technically solid in its implementation and benchmarking, but scientifically shallow in its control-theoretic grounding. It conflates high-level reasoning with genuine control synthesis and overstates LLM superiority without rigorous analysis or evidence against traditional methods in MIP or MIP with the integration of LLMs.

[1] Deits, Robin, and Russ Tedrake. "Computing large convex regions of obstacle-free space through semidefinite programming." Algorithmic Foundations of Robotics XI: Selected Contributions of the Eleventh International Workshop on the Algorithmic Foundations of Robotics. Cham: Springer International Publishing, 2015.
[2] Deits, Robin, and Russ Tedrake. "Efficient mixed-integer planning for UAVs in cluttered environments." 2015 IEEE international conference on robotics and automation (ICRA). IEEE, 2015.
[3] Manchester, Zachary, and Scott Kuindersma. "Variational contact-implicit trajectory optimization." Robotics Research: The 18th International Symposium ISRR. Cham: Springer International Publishing, 2019.
[4] Dantam, Neil T., et al. "Incremental task and motion planning: A constraint-based approach." Robotics: Science and systems. Vol. 12. 2016.
[5] Garrett, Caelan Reed, et al. "Integrated task and motion planning." Annual review of control, robotics, and autonomous systems 4.1 (2021): 265-293.
[6] Thomason, Wil, Marlin P. Strub, and Jonathan D. Gammell. "Task and motion informed trees (TMIT*): Almost-surely asymptotically optimal integrated task and motion planning." IEEE Robotics and Automation Letters 7.4 (2022): 11370-11377.

**Questions:**

1. You state “first to fine-tune LLMs for M3P … achieving 98.8% (single) and 89.6% (multi).” What, precisely, is novel beyond (i) text-to-MIP translation and (ii) exposing a discretization API? Please distinguish between novelty and prior LLM→MIP/modeling works, as well as LLM+P systems.
2. RoboM3P is 2D and API-scaffolded; what prevents overfitting to your API grammar rather than general geometric reasoning? Any tasks built from independently authored code (distribution shift)?
3. Can you expose intermediate-point confidence (or constraints) so the optimizer can repair poor waypoints automatically?
4. How do you detect/diagnose infeasible scripts before solver waste? Any LLM-driven repair loop (e.g., constraint relaxation suggestions) and guarantees to preserve safety constraints?

---

### Official Review · Reviewer_tFj6 · 2025-11-01

**Soundness:** 2
**Presentation:** 2
**Contribution:** 2
**Rating:** 2
**Confidence:** 4

**Summary:**

This paper introduces LLM+MIP, a large language model that can translate natural-language multi-modal motion planning problems into MIP scripts.

**Strengths:**

LLM+MIP, not LLM+P, is a good idea. A new research direction for robotics.

**Weaknesses:**

Please check the questions.

**Questions:**

Collision-free region construction, grasp wrench checks, piecewise-linear IK, and spatial side-constraints are encapsulated as high-level, LLM-callable APIs. When these APIs are removed, success drops to ~0%. It shows the approach doesn't rely on translation ability. The novelty of this paper is MIP for M3P with provided discretization APIs, not what is claimed in the paper.



The paper studies multi-mode planning, not multi-modal inputs.



All experiments are 2D. Please add dynamics and 3D evaluation at least.



Success metrics mixed solvable code and correct semantics.



Add PDDLStream, PyTAMP, RRT* with mode inference, TrajOpt, BIT*, hand-written MIP formulations by experts, and LLM+P followed by geometric layers as baselines.



Add details about template diversity, splits, and how held-out tasks avoid overlap in constraints and composition patterns.



The paper missed the core research challenge, which is choosing discretization granularity and assessing fidelity vs. tractability. Please add results on LLMs designing discretizations or adaptively refining them.

---

### Official Review · Reviewer_mctK · 2025-11-01

**Soundness:** 2
**Presentation:** 3
**Contribution:** 2
**Rating:** 4
**Confidence:** 4

**Summary:**

The paper proposes an LLM-guided pipeline that translates natural-language multi-modal motion-planning (M3P) problems into approximate mixed-integer optimization programs. An LLM generates CVXPY code augmented with a library of LLM-callable discretization APIs so that non-convex constraints are approximated as unions of convex sets solvable by MIP solvers.The authors introduce RoboM3P, a 1.5k-instance dataset pairing task text and obstacle maps with reference MIP scripts. The dataset includes five single-domain tasks and seven multi-domain compositions. Using this benchmark, they compare in-context learning to supervised fine-tuning. The best model reports 98.8% success on single-domain and 89.6% on multi-domain tasks.
A decomposed strategy that predicts feasible waypoints and solves smaller sub-MIPs is also studied.

**Strengths:**

- The authors propose a translation of motion planning problems to mixed-integer optimization programs which is novel.

- The paper introduces the RoboM3P dataset which contains pairings of  natural-language tasks and maps with executable MIP code as structured API calls. The dataset is useful for the research community.

- With SFT-Full, the best model reaches 100% formulation correctness and high success rates. It shows substantial and consistent improvements over ICL and smaller models.

- Sections detailing the dataset components and API categories make the pipeline easy to follow. Code is represented as function-like API calls, improving readability and reproducibility.

**Weaknesses:**

1) The scope is narrow and largely composed of toy tasks. The multi-domain compositions are limited to at most two tasks. This undermines claims about general impact.

2) The method’s framing as LLM-driven “mathematical reasoning” may overstate the model’s role. In practice, it appears to rely heavily on a hand-crafted API. The LLM assembles prebuilt primitives rather than deriving new constraints. The ablation suggests this dependency on the prebuilt primitives is central to the success of the framework. The performance drops sharply without the complex APIs which indicates limited evidence of genuine mathematical understanding. Overall, the system seems better characterized as an effective orchestrator of expert defined functions than a general reasoning engine.

3) The LLM+P (PDDL) and classical TAMP stacks are missing as baselines. Prior work uses LLM to PDDL with planners. The paper motivates LLM to MIP against that line but does not compare quantitatively.

4) “Formulation-Correctness,” “Parameter-Correctness,” and “Success-Rate” are defined, but success is considered when “a feasible plan produced by a solver,” with no optimality/quality/robustness criterion or tolerance. Moreover, there isn’t an evaluation metric that determines the success of the overall pipeline.

5) The method is primarily learning to use a provided library of discretization primitives rather than independently deriving complex constraints. The approach is therefore powerful within the API and task distribution it’s finetuned on, but may be brittle to changes in the primitive set or to distribution shifts that weren’t covered by training. The ablation indicates that removing complex APIs drives success to zero.

If the analysis is significantly extended and the main concerns brought up here are clarified, the value proposition of the paper would be greatly increased.

**Questions:**

In addition to the remarks made above (Weaknesses), here are some further questions:
1) How does your approach compare to LLM→PDDL or a classical TAMP pipeline on the same instances in terms of compute times and the success rates?

2) Does the model demonstrate mathematical reasoning beyond memorizing API usage patterns? Please report a leave-primitives-out evaluation: withhold a subset of primitives (and tasks involving them) during training, then test on tasks that require those withheld primitives or their novel compositions. How does performance and error profile compare to the in-distribution setting?

---

### Note · Authors · 2025-11-21

I have read and agree with the venue's withdrawal policy on behalf of myself and my co-authors.